# Efficiently accelerated free electrons by metallic laser accelerator

Dingguo Zheng [1,2], Siyuan Huang [1,2], Jun Li[1], Yuan Tian[1,2], Yongzhao Zhang[1,2], Zhongwen Li[1], Huanfang Tian[1], Huaixin Yang[1,2,3] & Jianqi Li [1,2,3,4] ✉

Strong electron-photon interactions occurring in a dielectric laser accelerator provide the potential for development of a compact electron accelerator. Theoretically, metallic materials exhibiting notable surface plasmon-field enhancements can possibly generate a high electron acceleration capability. Here, we present a design for metallic material-based on-chip laser-driven accelerators that show a remarkable electron acceleration capability, as demonstrated in ultrafast electron microscopy investigations. Under phase-matching conditions, efficient and continuous acceleration of free electrons on a periodic nanostructure can be achieved. Importantly, an asymmetric spectral structure in which the vast majority of the electrons are in the energy-gain states has been obtained by means of a periodic bowtie-structure accelerator. Due to the presence of surface plasmon enhancement and nonlinear optical effects, the maximum acceleration gradient can reach as high as 0.335 GeV/m. This demonstrates that metallic laser accelerator could provide a way to develop compact accelerators on chip.

Strong electron–photon (e–p) interactions, as the most important issue in laser-driven accelerators, have been studied for decades; for example, the inverse Smith–Purcell effect and relevant phenomena in metallic media have been researched[1,2]. In recent years, because of the high laser damage thresholds of dielectric materials[3–5], dielectric laser accelerators (DLAs) have been studied in many laboratories[6–8]. To achieve high acceleration gradients, various nanostructures (e.g., dual pillar gratings, distributed Bragg reflectors)[9–12] have been designed and many new techniques (including the pulse-front-tilted laser and multiple driving)[13–15] have been developed. Though the driving fields have reached 9 GV/m for the DLA[16,17], yielding the highest average acceleration gradient to date, the ratio between the average acceleration gradient and the incident laser field is still low. The exploration of high acceleration gradient accelerators and new laser-driven accelerator designs continues.

Metallic materials usually have a lower laser damage threshold than dielectric materials[4,18,19]. For example, the threshold of the gold grating is ~0.4 J cm$^{-2}$ with a temporal laser width of 1 ps (the value for fused silicon is ~3.5 J cm$^{-2}$)[3,4]. However, it should be also noted that the difference in laser damage thresholds between metallic and dielectric materials can be lowered by reducing the temporal widths of pulsed laser[4,5,18], and the high reflectivity of metal within specific wavelength ranges can also increase the damage threshold. Recent studies of surface plasmons in metals showed that the plasmonic near-field can be confined in a nanoscale space and enhanced by several orders of magnitude[20]. Theoretical studies have also reported that the surface plasmons on metallic metasurfaces designed for continuous electron acceleration can provide high acceleration gradients that are much greater than that of the incident optical field[21–23]. These facts indicate that metallic materials can also serve as good media for designing laser-driven accelerators.

The development of the ultrafast transmission electron microscope (UTEM) has provided a platform to investigate the nature of e–p interactions with nanometre-femtosecond (nm·fs)-

[1]Beijing National Laboratory for Condensed Matter Physics, Institute of Physics, Chinese Academy of Sciences, 100190 Beijing, China. [2]School of Physical Sciences, University of Chinese Academy of Sciences, 100049 Beijing, China. [3]Songshan Lake Materials Laboratory, 523808 Dongguan, Guangdong, China. [4]Department of Electrical and Electronic Engineering, Southern University of Science and Technology, 518055 Shenzhen, Guangdong, China. ✉e-mail: ljq@iphy.ac.cn

scale resolution[24–27]. For example, photon-induced near-field electron microscopy (PINEM) has attracted considerable attention because of its ability to image near-field distributions on the nanoscale and provide quantized energy information of electrons[28]. Recently, quantized electron energy peaks from DLA have been observed using UTEM[29].

In this study, metallic laser accelerators (MLAs) made from silver crystals, which are strong e–p interaction media, have been studied using the UTEM. Importantly, asymmetric features in electron energy-loss spectroscopy (EELS) spectra, in which most of the electrons lie in energy-gain states, have been observed in a bowtie nanostructure MLA; this natural characteristic could be applied to improve the on-chip acceleration performance and enable further study of quantum energy state modulation.

## Results

An experimental schematic diagram is shown in Fig. 1a, in which focused and collimated electrons graze across the MLA surface meanwhile it is illuminated by the pulsed femtosecond laser. The zero-loss EELS peak in the experiments is shown in Fig. 1b. It should be noted that all spectra in this paper are the electron energy-loss spectra, in which the right part of the plots is energy loss and the left is energy gain. In our measurements, the key experimental parameters are listed as follows: the primary electron energy center is 200 keV with full width at half maximum (FWHM) of ~2.4 eV; the electron beam size is ~45 nm (FWHM) and its center stays 20 nm (i.e., impact factor) away from the MLA surface; the electron pulse has a temporal width of 350 fs as measured by PINEM[25]. The pump laser is always centered at the wavelength of 515 nm with a repetition rate of 200 kHz, spot diameter of ~50 μm (FWHM), temporal width of 200 fs and fluence of 25.5 mJ cm$^{-2}$. Therefore, the laser intensity and driving electric field can be calculated to be $1.27 \times 10^{11}$ W cm$^{-2}$ and 0.978 GV/m, respectively.

To study the e–p interactions and the relevant plasmonic fields in metallic nanostructures, previously, we have performed an

investigation on a silver nanowire, which can be approximated as an MLA with only one period. It was observed that the electron energy gain for a single silver nanowire with a diameter of ~100 nm can be as high as 150 eV or more, as shown in Fig. S1; this illustrates the great potential for metallic nanostructures used in the development of laser-driven accelerators. Under laser irradiation, the responding of free electrons in the metal could lead to the redistribution of the electromagnetic field, allowing metallic material to focus and confine electromagnetic waves on local areas and generate strong near fields. To take advantage of plasmonic field enhancement, the structural period of the MLA should be decided by the phase-matching condition[2]: $\Lambda = \beta\lambda$, where $\Lambda$ is the spatial period, $\lambda$ is the wavelength of the incident light, and $\beta = v_e/c$ (where $v_e$ is the speed of the electron beam and $c$ is the speed of light in vacuum). A typical image of an MLA with 31 parallel teeth is shown in Fig. 1c. In our experimental measurements, the MLA's structural period was designed to be 358 nm based on theoretical calculations. A comparison of the MLA images taken before and after laser illuminations is shown in Fig. S2.

The spatial distributions of the accelerating field ($E_z$) at two typical points in time are shown in Fig. 1d and the deflection fields ($E_x$ and $B_y$) are shown in Fig. S3. Because of the occurrence of plasmonic field enhancement in the metallic nanostructure, the amplitude of the electric field on the nearby surface can be four times larger than that of the incident optical field. It is obvious that the laser can be reflected efficiently to form a standing wave, but this type of standing wave cannot contribute to the electron acceleration (see SI 1). The z-component of the evanescent wave can be expressed as follows (also see SI 1):

$$E_z(x,z,t) = E_0 \sum_n a_n e^{i(n\frac{k_0}{\beta}z - \omega t)} e^{-\frac{x}{\delta_n}}, \qquad (1)$$

where $E_0$ is the amplitude of the incident light, $k_0$ is the wavevector length in a vacuum, $\omega$ is the angular frequency, $a_n$ is the $n$th Fourier

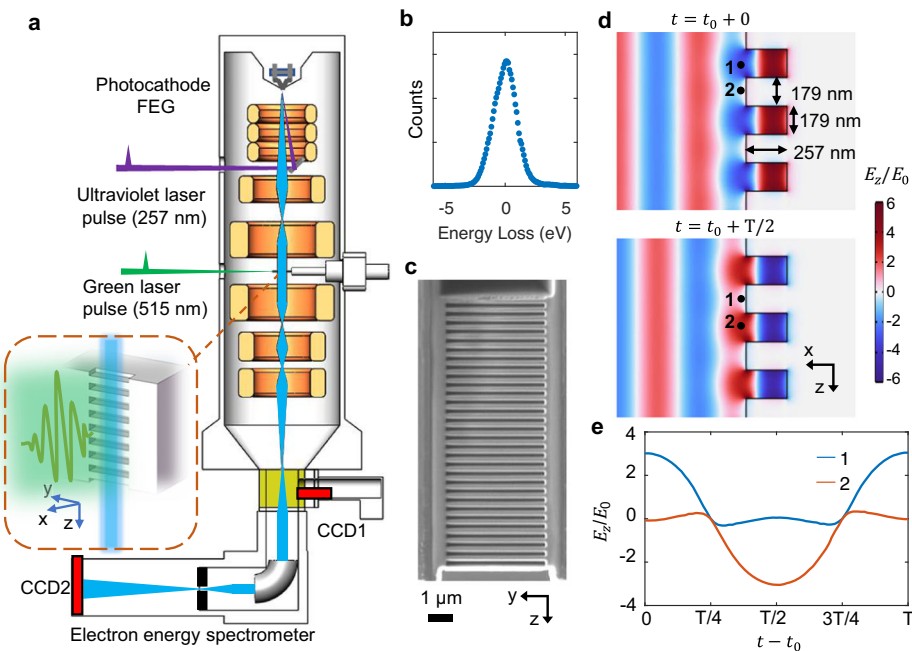

**Fig. 1 | Electron–photon interaction on an MLA. a** Schematic diagram illustrating the MLA in the UTEM system. The pump laser beam is perpendicular to the electron beam direction and laser polarization is parallel to the electron beam. **b** Initial electron spectrum in experiments, the electron beam energy is centered at 200 keV with FWHM of 2.4 eV. **c** A scanning electron microscopy image of the MLA with 31 periods (length 11.1 μm). Its structural parameters are shown in (**d**). **d** Simulated results of time-dependent electric field distribution. $T$ is the optical period, being the time of electron moves through one MLA period. The dots 1 and 2 represented two electrons with an impact parameter of 20 nm. **e** Time-dependent z-component of electric field experienced by electrons 1 and 2.

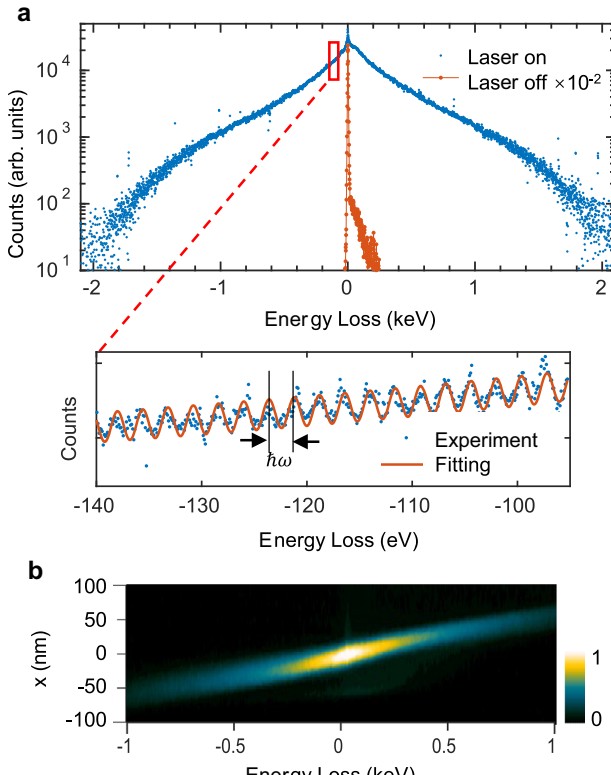

**a** Experimental spectra with spectrometer dispersion 1 (upper) and 0.1 (lower) eV per channel. The orange sideband from 2 to 250 eV arises from the interaction between electrons and the silver nanostructure with no laser excitation. **b** Relative electron count distribution image in the energy range of −1 to 1 keV.

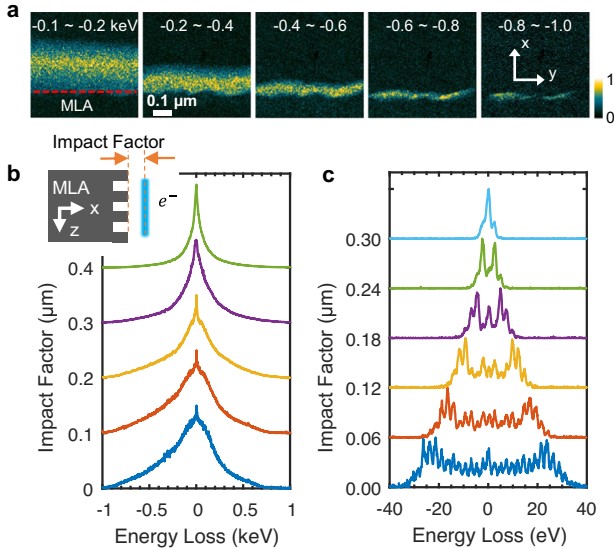

**Fig. 3 | Space-energy distribution of electrons after interaction with evanescent wave. a** A series of energy-filtered images show the relative count spatial distribution of energy-gain electrons. Corresponding energy windows are labeled in each image. The red dash line in the inset indicates the boundary of MLA. **b** A series of spectra obtained by moving electron spots away from the MLA surface. **c** A series of spectra obtained with the lower laser fluence (0.2 mJ/cm²) and wider laser pulse duration (~800 fs) compared with those of **b**, for better visualization of the quantized peaks.

**Fig. 2 | Acceleration results of an MLA. a** Experimental spectra with spectrometer dispersion 1 (upper) and 0.1 (lower) eV per channel. The orange sideband from 2 to 250 eV arises from the interaction between electrons and the silver nanostructure with no laser excitation. **b** Relative electron count distribution image in the energy range of −1 to 1 keV.

coefficient, and $\delta_n$ is the decay length of the $n$th Fourier expanded field. This type of evanescent wave can couple efficiently with an electron beam, as shown in Fig. 1e. When the $z$-component of the optical field nearby the MLA surface goes along the reverse direction of the $z$-axis, electron 1 can be accelerated. In the next half of the period, electron 2 can be decelerated by the optical field. Upon the phase-matching condition being achieved, electron 1 almost always experiences accelerating fields, while electron 2 almost always experiences decelerating fields. In fact, the existence of periodic acceleration and deceleration in the present case could compress a femtosecond-width electron pulse into a series of attosecond pulses after a distance of free-space propagation[30].

The experimental results for electron acceleration are shown in Fig. 2a. To generate strong acceleration fields, the optical polarization is set to be perpendicular to the teeth. Efficiently accelerated free electrons with the highest energy gain of 2 keV and quantized satellite peaks can be observed clearly when using suitable spectrometer dispersion. After grazing over the MLA, the free electrons also have visible changes in the $x$-direction, as shown in Fig. 2b, which clearly illustrates the scattered electron count distribution; it is noticeable that electrons with higher energy gains also gain comparatively larger transverse movements.

The evanescent wave in Eq. (1) has an acceleration gradient of $G = eE_0 a_1 e^{i\phi_0} e^{-\frac{x}{\delta_1}}$ (see SI 1). Here, $\phi_0$ is the phase of the plasmonic near-field when the electron enters the accelerator. This formula can explain the experimental results in the upper panel of Fig. 2a well, but it cannot explain the separated peaks shown in the lower part of Fig. 2a. Instead, PINEM theory[31–34] can explain the quantized electron energy distribution well by introducing a dimensionless parameter $g$ (which is represented by $\beta$ in some entries in the literature) to describe the coupling strength; this is also discussed in the SI 2.1. Ignoring the

dephasing from the electron energy variation during the interaction, the relationship of $g$ to $G$ can be written as

$$|g| = \frac{N\Lambda}{2\hbar\omega}|G|, \qquad (2)$$

where $N$ is the number of periods and $\hbar$ is the reduced Planck constant. In the experimental results shown in Fig. 2a, $|G|$ could reach up to 0.18 GeV/m (interaction distance 11.1 μm) and $|g|$ is ~417.

Figure 3a includes a series of energy-filtered images with different energy windows, showing the energy-gain electron distributions near the surface of the MLA. The counts of energy-gain electrons show an increase and gradually decay along the $x$ direction, and the high energy-gain electrons can only be observed in the areas very close to the surface. Figure 3b displays the impact factor-dependent electron energy spectra and Fig. 3c shows the corresponding data obtained under low laser fluence and wide laser pulse duration. The resulting electrons, after interacting with tens of periods of near fields, spread into a few or hundreds of energy states. Additionally, asymmetric spectra can often be observed when the electrons are very close to the MLA surface, as shown in the bottom electron spectrum in Fig. 3b; this will be discussed extensively in the following context.

In previous studies, the propagation and focusing of surface plasmons in nanostructures have been investigated extensively as main issues and it was noted that the bowtie structure could greatly enhance the local fields by orders of magnitude upon fs-laser excitation[20,35,36]. A bowtie-structured MLA was therefore designed to accelerate electrons (Fig. 4a). Simulation results demonstrate that strong near-fields appear in the gap between the two triangles and $E_z$ can be enhanced by more than 100 times close to the tip areas, as shown in Fig. 4b. Our typical experimental spectra, which demonstrate that strong e−p interactions did indeed occur in this bowtie-structured MLA, are shown in Fig. 4c. The highest energy-gain could be as much as 2.4 keV, indicating that an electron could absorb 1000 photons in the

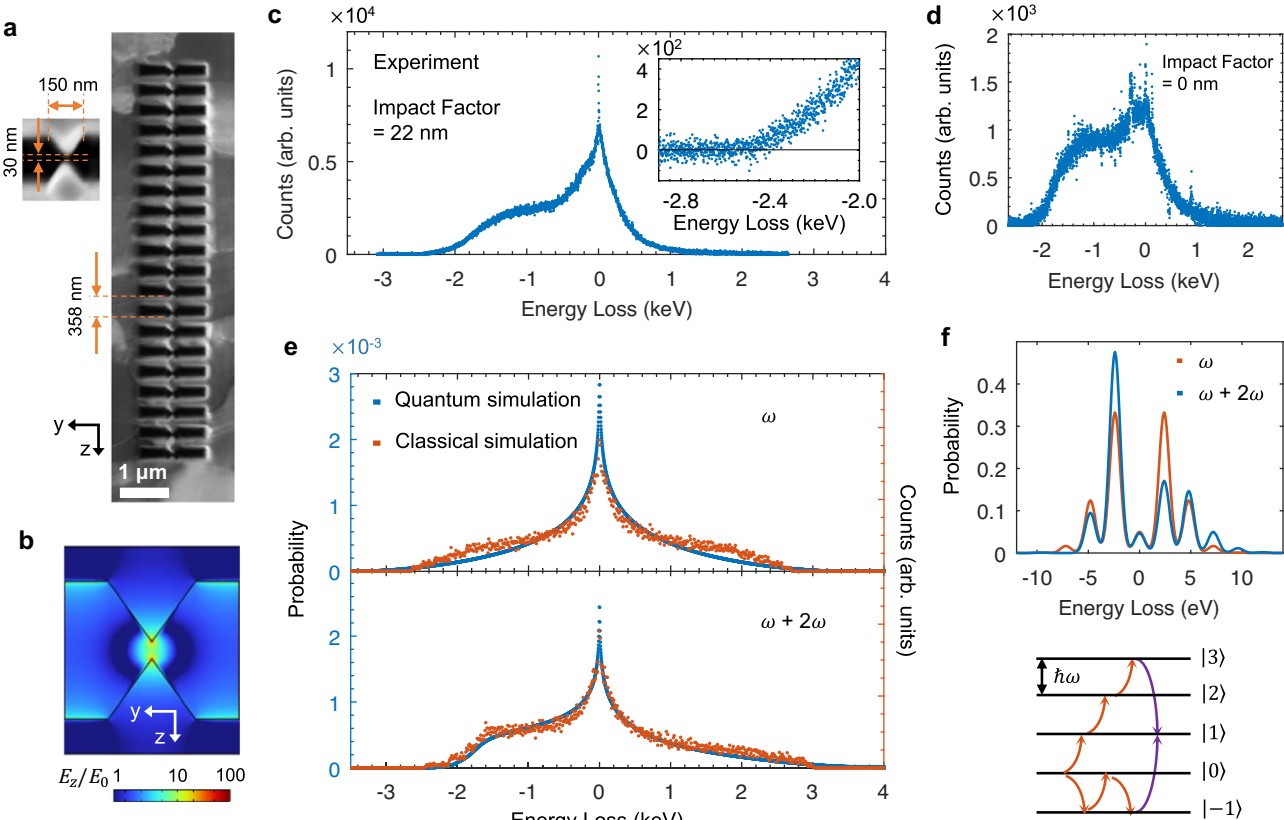

**Fig. 4 | Acceleration on a bowtie structure MLA. a** Scanning electron microscopy images demonstrate tip-to-tip triangles (left) applied in MLA and a sample (right) with 20 periods used in experimental measurements. **b** Simulation for field enhancement in one period within the $x = 0$ plane (i.e., the metal–vacuum interface). **c** Experimental electron spectrum obtained in the bowtie MLA. The inserted figure shows a close-up of the acceleration tail. **d** The same to **c** but the electron beam is very close to the MLA surface. **e** Electron counting distribution obtained from particle tracing simulation and electron probabilities on the energy states obtained from quantum theory simulation for only the fundamental plasmon field (up panel), as well as for coexisting with the SH field (low panel). **f** Theoretical probabilities of an electron in each energy state at low field intensity ($|g_1| = 1, |g_2| = 0.2, \delta = 0$) and the corresponding diagram that shows the typical transition paths in energy levels yielding the appearance of high probabilities in energy-gain states.

7.16 μm interaction length. Table S1 lists experimental data to illustrate a comparison between bowtie-structured MLAs and DLAs.

The MLA acceleration gradient could reach 0.335 GeV/m, i.e., ~1/3 of the incident field. So far, this value is the highest record of average acceleration gradient in the single grating structure-based laser-driven accelerator[16], this fact suggests that the prototype MLA could have high performance in the development of compact accelerators. Furthermore, another significant feature shown in Fig. 4c is that most electrons (up to 76%) lie in energy-gain quantum states, which is rather different from the conventional spectra obtained in studies using PINEM[28,29] or DLAs[6–9,37]. We attribute the asymmetric structure in Figs. 3d and 4c, d to optical nonlinear effects, i.e., the second-harmonic (SH) field with wavelength 257 nm generated on MLA induced by the incident field. The occurrence of high-harmonic generation in this bowtie structure has been demonstrated by optical methods in previous works[35,36,38]. The spatial distribution of the SH fields in a single silver nanowire after the fs-pulsed laser excitation has been directly observed in real space by PINEM[39]. This asymmetric feature of the electron spectrum is also consistent with the PINEM experimental measurements of two laser beam excitations (one with frequencies $\omega$ and another with $2\omega$)[40]. Furthermore, the SH field would cause asymmetric features in the EELS spectrum of the PINEM experiment, as predicted theoretically by Konečná et al.[41]

In order to understand asymmetric features in the electron energy distribution, we have performed a particle tracing simulation. Considering the cases of only fundamental plasmon field and coexistent with SH field, we can obtain simulation results for electrons passing 20 periods of the MLA as shown in Fig. 4e. These results are in good

agreement with our experimental data, in which the appearance of asymmetric electron energy distribution is evidently obtained arising from the SH field. To clearly show that the asymmetry is not due to partial electrons hitting the nanostructure and uncollected by electron spectrometer, another simulation is carried out. When the incident electron beam is very close to MLA (small impact factor), some electrons will hit MLA (Fig. S4), but the final electron energy spectrum is symmetric.

To address the influence of multimode fields on the electron wavefunction and quantum energy states, the quantum theory of the interaction of electrons with multimode fields is presented in SI 2.2. After passing through multimode fields, the electron wavefunction can be expressed as

$$\Psi(z,t) = e^{i(p_0 z - E_{e,0} t)/\hbar} \psi_0(z - v_e t) \prod_j \phi_j(z,t),\tag{3}$$

where $\psi_0(z - v_e t)$ is the initial envelope function and $\phi_j(z,t)$ contains the information for the $j$th mode field. This equation indicates that the effect of multimode fields on the wavefunction can be understood as multiplications by the modulation functions of each mode field.

If only the fundamental field and the SH field exist, the probability of an electron gain $n\hbar\omega$ is given by

$$P_n(|g_1|, |g_2|, \delta) = \sum_{l=-\infty}^{\infty} \sum_{m=-\infty}^{\infty} \alpha_{lm} \exp(-i(l-m)\delta),\tag{4}$$

where $g_1$ and $g_2$ represent the coupling parameters of the fundamental field and the SH field, respectively, $\delta = 2\arg(g_1) - \arg(g_2)$ is the relative phase of the fundamental and SH fields, and $\alpha_{lm} = J_{n-2m}(2|g_1|)J_{n-2l}(2|g_1|)J_m(2|g_2|)J_l(2|g_2|)$, where $J_l$ is an $l$th-order first-kind Bessel function. Figure S5 illustrates the intensity dependence and the relative phase effects on the probability of an electron in each energy state. Noticeably, the SH field will result in an asymmetric distribution and yield a higher probability of an electron in an energy-gain (loss) state when $\delta = 0$ ($\delta = \pi$).

If the SH field and strong e−p interactions are considered (the weighted average $|g_1| = 233$, $|g_2| = 23.3$, $\delta = 0$), theoretical calculations are shown in Fig. 4e. Detailed data of simulation is shown in Fig. S6 and SI 2.3. To demonstrate how these asymmetric structures are created, a typical energy state distribution with low field intensity and a typical transition diagram is presented in Fig. 4f. They showed that the existence of the SH field would induce electron transitions into specific states, resulting in an asymmetric energy state distribution. These quantum energy state modulations indicate that MLA can efficiently reduce the energy spread and pave the way for a generation of high-energy monochromatic electron beams via on-chip accelerators.

## Discussion

Metallic material-based laser-driven accelerators have been designed and studied using the UTEM, and the main features in correlation with e−p interactions have been characterized well via PINEM imaging and the EELS spectra. Benefiting from plasmonic field enhancement in the metallic nanostructure, efficient free electron acceleration can be achieved by MLA, with a high acceleration gradient comparable to that of DLAs. In an MLA with bowtie structures, asymmetric EELS spectra were observed, with the vast majority of the electrons in the energy-gain states. A prototype of the proposed MLA provides a way to realize compact accelerators on chips, but it should be noted that dielectric materials generally have higher laser damage thresholds.

## Methods

### Experimental details

The MLAs were made from silver plates; they were fabricated using a focused ion beam (FIB) system (Helios NanoLab 600i DualBeam, FEI Inc.). Firstly, using a high current ion beam to create a flat cross-section on a silver plate. Then the surface was polished in advance by low low-current ion beam. Secondly, setting the FIB in matrix etched mode with a current of 7.7 pA and a voltage of 5 kV. For the simple grating structure, the matrix unit is set to a rectangle. For the bowtie structure, the matrix unit is composed of two rectangles and two tip-to-tip triangles. Thirdly, polishing the etched surface again. This step is necessary because the scattering and deposition of Ag ion and Ga ion would make the surface rough again. Finally, moving the silver plate to a modified TEM sample holder.

Experimental measurements were performed on our in-house-built UTEM, which was constructed by modifying a commercial Schottky-type field-emission microscope (2100F microscope, JEOL Inc., Tokyo, Japan). Details of this UTEM were reported in one of our previous papers[25]. The initial pulsed laser beam with a centre wavelength of 1030 nm and a temporal pulse duration of 200 fs was split into two laser beams using a β-barium borate crystal. A frequency-doubled laser (515 nm) was introduced to the specimen room for pumping and a frequency-quadrupled laser (257 nm) was introduced to the field emission gun to create the photoelectrons used in stroboscopic mode. The pump laser's polarization was controlled using a half-wave plate. During these experiments, the UTEM system was operated with a repetition rate of 200 kHz and less than one electron per pulse (single-electron mode) to avoid the space-charge effect. A post-column Gatan spectrometer (GIF 965) with a 2048 × 2048 pixel charge-coupled device (CCD) camera was used to record the spectra with a typical exposure time of 100 s (shown in Fig. 1a as CCD2). The energy-filtered images were recorded using a 256 × 256 pixel CCD sensor with a typical exposure time of 80 s. For the wide energy spread spectra (>1 keV), the spectrometer dispersion was 1 eV per channel with a total channel number of 2048.

### Theoretical simulation

Theoretical calculations were performed using commercial finite-element software (COMSOL Multiphysics 6.0; radio frequency module). The incident electromagnetic wave was modelled as a linearly polarized plane wave.

In a two-dimensional space, the maximum element size of the mesh was set at 20 nm. The Drude−Lorentz dispersion model was selected as the electric displacement model in the wavefunction. The related high-frequency permittivity and plasma frequency data from the literature[42] were used in this model. Figure 4b was calculated to characterize the features of the surface near-fields in three dimensions; the maximum element size of the mesh was set at 4 nm in the area very close to the tip.

In the particle tracing simulation, 10,000 electrons with an impact factor from 0 to 45 nm are taken into our calculation. Both the time-dependent acceleration ($eE_z$) and time-dependent deflection force ($eE_x + ev_e B_y$) are considered for the electron grazing over the 20-period-MLA. In the case of coexisting $\omega$ and $2\omega$ fields, two light beams with wavelengths 515 and 257.5 nm are incident to MLA. Only electrons without hitting the MLA are counted in the final electron energy spectra.

## Data availability

The data of all two-dimensional plots (e.g., electron spectra) and most three-dimensional plots (e.g., electron images) generated in this study have been deposited in the figshare database under the accession link (https://doi.org/10.6084/m9.figshare.23937495). A few large-size auxiliary images within the article and the data in supplementary information are available from the corresponding author upon request.

## Code availability

Numerical simulations in this work are performed using the commercial finite element software COMSOL Multiphysics. All related codes are available from the corresponding authors upon request.

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

## Acknowledgements

This work was supported by the National Natural Science Foundation of China (Grant Nos. U22A6005 , 12074408, 52271195), the National Key Research and Development Program of China (Grant No. 2021YFA13011502), the Strategic Priority Research Program (B) of the Chinese Academy of Sciences (Grant Nos. XDB25000000, XDB33000000), the Scientific Instrument Developing Project of the Chinese Academy of Sciences (Grant Nos. YJKYYQ20200055, ZDKYYQ2017002, and 22017BA10), the Synergetic Extreme Condition User Facility (SECUF), the Guangdong Major Scientific Research Project (2018KZDXM061), Beijing Municipal Science and Technology major project (Z201100001820006) and IOP Hundred Talents Program (Y9K5051).

## Author contributions

Conceptualization: D.G.Z., J.Q.L. Methodology: D.G.Z., J.L., Z.W.L., Y.Z.Z. Investigation: D.G.Z., Y.T. Visualization: D.G.Z., S.Y.H. Instruction: J.Q.L., H.F.T., H.X.Y. The manuscript was written through the contributions of all authors. All authors have given approval to the final version of the manuscript.

## Competing interests

The authors declare no competing interests.
