## [Peer Review File · Nature Communications]

Efficiently accelerated free electrons by metallic laser acceleratorREVIEWER COMMENTS

Reviewer #1 (Remarks to the Author):

The authors present an experimental study of a plasmonic metasurface laser accelerator. They show experimental studies involving two types of MLAs – “conventional” gratings and a periodic array of bowtie antennas, exhibiting acceleration gradient of up to 180MeV/m and 335MeV/m respectively. In additions, the authors discuss the possibility to excite the MLA with a field comprising a superposition of two harmonics, showing that such approach can lead to asymmetric features in the EELS spectrum which can be controlled by the phase shift between the two frequencies.

As a general comment, I believe that this is an important work. To my knowledge this is the first experimental demonstration of an MLA. Furthermore, that fact that this first experiment exhibits normalized acceleration gradient (or “efficiency” as the authors call it) which is in the same range as the best DLA result, indicate that the MLA approach could be advantageous for applications where high pump power are inapplicable. In that context, I believe the manuscript should be published pending the modifications and the revisions detailed below.

Most of the manuscript is clearly written and well explained. However, there are a few less-clear sections that should be improved for the benefit of the reader. Specifically, the section dealing with the inelastic scattering of free electrons by an incident laser is not clear. Figure 4c is also puzzling – I found it quite difficult to understand the authors meaning and what the process presented in this figure is.

There are several missing pieces of information which must be included:

1. In Fig. 1 the authors show the calculated field in the direction of the acceleration. The amplitude of the field in the orthogonal direction (perpendicular to the MLA) should also be presented and discussed. Such field induces deflection forces that can modify the particle height above the surface and, hence, the average acceleration gradient
2. The manuscript lacks a comprehensive theoretical analysis of the expected energy gain/loss spectrum. Specifically, it is not clear whether the theoretical analysis in Fig. 5 is based on a complete particle tracking simulation, 1D relativistic “Newton-law” based calculation, etc. It is also unclear whether the theoretical analysis assumes a specific height of the electron beam above the MLA or does it take into consideration the distribution of the electrons in the E-beam. Are the perpendicular deflection forces considered in the model? Do the authors consider particles that hit the metallic structure or electrons which move away from the MLA and do not reach the detector?
3. Some technical details on the laser and electron beams are missing. What is the size of the E-beam? Where is it aimed at with respect to the MLA (height above the surface)? What is the amplitude of the driving field (this piece of information is hiding in the SI – it should be clearly indicated in the main text)?
4. There is an apparent “discrepancy” in the choice of the grating period. Assuming $\beta=0.7$ and $\lambda=515\text{nm}$ the periodicity should be 361nm and not 358nm. To resolve this, one needs to calculate the precise β of the electrons (0.695 for 200keV) to find that the period should be 357.8nm –

corresponding to the choice of 358nm of the authors. I suggest the authors use the precise numbers to avoid confusion.

5. Referring to Fig. 5c, the zoom-in inset does not show the photon energy quantization which is expected according to inset of Fig. 5d. The authors should explain why they are unable to show this, particularly in view of the high-resolution EELS their apparatus exhibits, as seen in Fig. 2a.

6. Referring the same figure, it is not clear what the fundamental and second harmonic are. Is it 515nm and 258nm or 1030nm and 515nm? Furthermore, what are the intensities of the two beams?

7. With respect to comments 5 and 6, the claim at the end of the section dealing with Bowtie structure MLA – “They showed that the existence of the SH field would induce transitions into specific states, resulting in an asymmetric energy state distribution”, is not well-supported by the experimental results.

8. There is a typo referring to Fig. 1f (lines 134, 135, 141). I assume the meaning is Fig. 2a.

Finally, one of the great promises of MLA is their alleged ability to provide “efficiency” that exceeds unity (i.e. obtaining normalized acceleration gradient which is larger than 1). Although the results presented by the authors are impressive, they still fall short compared to those of contemporary DLA structures. I expect the authors to provide a detailed discussion of the mechanisms limiting the acceleration gradient of their structures and what can be done in order to improve it. Does the reason for the limited gradient stem from the fact that the MLA is not tapered? Does it have to do with the choice of material (Ag)? The unit-cell structure (bowties and not, say slit antennas as has been suggested in previous theoretical studies)? etc.

Reviewer #2 (Remarks to the Author):

The authors describe their experiments of facilitating silver-based nanophotonic structures (antennas) to manipulate 200 keV electrons in an ultrafast transmission electron microscope, with the stated goal of having these structures (and in particular: metal) as an electron accelerator on-chip. Metals have many advantages over dielectrics for serving as electron accelerators, such as high conductivity and the ability to create smaller features (when discussing MLAs, as is the case of this work). However, there is one great disadvantage, which is the much lower damage threshold of metals versus dielectrics.

This manuscript presents well the authors’ experiments and is an important contribution to the on-chip nanophotonic electron accelerator community. However, there are two fundamental difficulties that must be addressed prior to publication (which I fully support! Great results!). These are namely: the damage threshold and nonlinear effects.

Below I list minor comments to be addressed by the authors, and conclude with the above main questions.

First, throughout the manuscript, the authors should differentiate between acceleration (GeV/m) and field strength (GV/m).

Abstract:

Mentions the acceleration gradient of 0.335 GeV/m, but should also reflect in a sentence the reason for this (maximum) gradient. What is the damage threshold?

Page 3:

- The authors write,

“However, the required acceleration gradient on the gigavolt per metre scale has not been realized by DLA to date, and the exploration of new laser-driven accelerator designs continues.”

In Cesar, *Comm. Phys.* 1 (2018) (<https://doi.org/10.1038/s42005-018-0047-y>) the authors demonstrated up to 9 GV/m driving fields with 1.8 GV/m in the accelerating mode. Due to technicalities in the design and experiment, the average gradient is 0.85 GeV/m. Although strictly speaking the average gradient is below 1 GeV/m, I would suggest to rephrase the above sentence to reflect the results of Cesar. In general, the authors and readership may find the recent DLA review paper in *Advances in Optics and Photonics* [<https://doi.org/10.1364/AOP.461142>] useful, as it contains a table of recent DLA experimental works.

- Still on page 3, “It should also be noted that the threshold gap between metallic and dielectric materials decreases” – what is a threshold gap? This should be corrected/the definition should be added to the manuscript.

Page 4:

- The authors mention an energy gain of 150 eV when interacting with a single silver nanowire. The laser wavelength (photon energy) and electron energy should be mentioned alongside this.

- Also, the authors make a sharp jump from the first few sentences about the silver nanowire to “Figure 1a shows.. MLA”. I would suggest to make a smoother transformation by adding one more sentence before the description of Figure 1. The details of the laser (pulse length, wavelength, photon energy, spot size on sample, intensity) should be mentioned here. One could start this section by a short description of all the experiments to be detailed in it.

Page 5:

- Figure 1c – add a scale bar.

- “but this type of standing wave cannot be attributed to electron acceleration (see the Supplementary Information (SI))” – please provide the correct section in the SI. This comment should be taken into account anywhere in the manuscript where a reference to the SI is made.

Page 6:

- “When the electron movement direction is opposite to the optical field direction near the MLA surface” – this is confusing: the electron trajectory is always towards +z. Do you mean to say, “when the polarity of the optical field is negative”? Should be rephrased.

- “In fact, periodic acceleration and deceleration of this type would compress a femtosecond-width electron pulse into a series of attosecond pulses²⁸” – actually, free-space propagation is required for this type of ballistic bunching, as shown in Ref. 28.

- “This formula can explain the experimental results in the upper panel of Fig. 1f well “ – there is no panel (f) in Figure 1?

- The authors define the relationship between g and G as dependent on the number of periods N (also derived in the SI). This has a hidden assumption, that there is no dephasing (in this periodic case - electron energy doesn't change during interaction). I suggest to add this remark, especially when making the calculation to $g=833$.

Page 7:

- Figure 2: it is not clear what “space” means in panel b. It is also imperative that the authors add a comparable measurement with laser off, both for panels (a) and (b), to stress the differences. What is the color scale in panel (b)?

Page 8:

- Figure 3: please use a line to mark the edge of the MLA in the panels, or alternatively add a panel with a “normal” (non-energy-filtered) image. What is the curious oscillation we see at the bottom of yellow-blue electron cloud?

- What does the y-axis “distance” in Fig.3b and c mean? Which distance?

- The authors promise to “discuss extensively” the asymmetric spectra in Fig.3b, however, I could not find such a discussion.

Pages 8-11 and Figure 4:

- I assume that at $\Delta t = -6.7\text{ps}$ the laser light and electrons do not interact (at all). Why are Fig.4a and Fig.4b (first row) not identical?

- Please add a line to depict the interface of the MLA in the panels.
- The explanation of Figure 4 is not very clear. It seems to me that when the laser is incident at an angle to the MLA, the incident and reflected beams generate transverse components of the electric field, which can explain the “X” shape. Is this kind of classical description correct?
- Similarly, the deflection (scattering) in the direction normal to the MLA surface (we see this particularly in the left column): could this be a result of the forces acting on electrons that are not at the peak of the electric field, and are so deflected “into” or “away” from the MLA surface?

Pages 11-14 and Figure 5:

- Page 11: “In our recent study” – did you mean, “in this work”? Otherwise, supply a reference.

This is the core result of this work: showing a nanophotonic antenna interact with an electron. The authors have two claims here:

(1) Nonlinear effects distort the spectrum towards energy-gain.

I understand from the authors that they assume their “fundamental” on the sample (i.e. 515nm light) goes through a nonlinear process and generates its second-harmonic (257nm). These two interact and are responsible for the energy-spectrum distortion of the electrons. I see some issues with this claim, which to me makes this claim tough to accept:

What is the efficiency of the SH-generation? What would be the calculated field amplitudes (in GV/m) for each?

What is the decay length from the antenna for 515nm, as opposed to the decay length for 257nm?

How does the doubled wavelength (257nm) impact the phase-matching condition with the electrons?

What is the efficiency of coupling of the fundamental (515nm) versus the SH (257nm)?

What is the absorption in Silver for these two wavelengths?

Why is it true that the energy distortion is towards the energy gain, rather than loss? Why isn't it a symmetric distortion? How can you experimentally control whether the distortion is towards this or that direction?

Were Figure 5d and Figure S6 based on theoretical calculations or a full numerical COMSOL simulation, including the SH generation?

I see the authors' claim with Figure S6 (which is a simulation, if I'm not mistaken), and the calculations, but can you prove (at least by answering the questions above) that this is indeed the case in the experiment? It might be that you can indeed get this result from a combination of 515nm+257nm, but I am not convinced yet that this is the case. I suggest to look at particle-tracking simulations (please include in the SI) and see if e.g. more of the decelerating particles “crash” into the structure, leaving a higher number of accelerated particles to reach the energy analyzer. The angle of divergence might also

affect the energy analyzer in experiment, by misinterpreting high angles with energy shifts (what aperture did you use?) or missing the energy analyzer altogether.

(2) Damage threshold of Silver in the experiment.

The technical details the authors provide did not convince me that their structures can withstand 1 GV/m incident fields. As a result, I am not convinced that the acceleration is indeed 335 MeV/m.

On page 11, the authors write:

“The MLA acceleration gradient could reach 0.335 GV/m, i.e., approximately 1/3 of the incident field. This indicates that the prototype MLA can generate an acceleration gradient and efficiency comparable with that of the highest performance DLA6,9,43,44.”

First, note again the work by Cesar with 9 GV/m fields. This claim should be changed.

How did you calculate “approximately 1/3 of the incident field”? Please convince me by adding a section to the manuscript (or SI if you must, but the expert readers would appreciate it in the manuscript) with a complete detailed list of your laser parameters (average power, beam size in x,y, repetition rate, pulse length) on the sample (i.e.: your second-harmonic = 515nm), and the formula you used to convert this to the peak optical field in GV/m. Also mention the peak intensity and fluence. Please also mention the electron pulse length (FWHM) and beam size (FWHM) and how you measured each.

Photonic nanostructures have a different damage threshold than flat surfaces. The authors have surely burned more than one of their structures. It would only strengthen the authors’ claim if they add a section to the SI showing several images of the structures after laser irradiation, comparing before- and after-damage and showing the types of damage they see, and at which laser parameters.

How did the authors deduce an acceleration to 2.4 keV? It’s most important to be very careful here. Figure 5c shows this, but I would like to see a close-up on that acceleration tail. What is the noise level of this measurement? Can you assert that you are above the noise level and, that this feature is indeed a “real” measurement and not an effect of aberrations through the spectrometer (e.g. high-angle spatial aberrations translate to energy)? A corresponding measurement of the spectrum super-imposed on this curve, with laser off but otherwise exact same parameters, would help quite a lot here.

In the conclusion, the authors write “A prototype of the proposed MLA provides a new way to realize compact accelerators on chip.” I agree: MLAs have great advantages over dielectrics. But, the authors should still remind the reader of the damage threshold: a conclusion from this work is still, that dielectric materials have been experimentally shown to endure at least 1 order of magnitude larger laser fields.

Reviewer #3 (Remarks to the Author):

In this paper, the authors present experimental demonstration of electron acceleration in the vacuum region above a microstructured device excited by an optical laser pulse. Two structures are considered: one is a simple metal grating and the other is a sequence of “bow-tie” structures arranged in a periodic 2D array on a planar substrate. The authors also make some interesting observations about transverse deflection of the particles under interaction with the laser in vacuum. In recent years, a variety of related experiments have been conducted by various researchers, primarily using structures made of silicon or silicon dioxide. Metals have been generally avoided due to damage limits of the laser. The authors here point out that some metals can have damage thresholds comparable to silicon under certain laser parameter conditions and thus can provide similar levels of “accelerating gradient” to a charged particle. In addition a plasmonic enhancement effect can potentially allow for more efficient excitation of the accelerating field. The authors include a comparison of their results against some of this prior art. While some aspects of the work appear repetitive of experiments conducted by previous researchers, the most novel and interesting aspect presented in this paper is the use of the metallic bowtie structure. The novel aspects merit publication, and the general treatment of the subject matter seems appropriate for Nature Communications. However, the structure and wording of the paper is often confusing and I believe requires substantial improvement in organization, choice and placement of figures, and the manner in which the results are explained and motivated.

I outline below various points that I feel should be addressed by the authors.

1. The abstract emphasizes the so-called bowtie structure as the primary result. However, the authors spend a substantial part of the paper initially describing various other results, including interaction with a nanowire and with a metallic grating, quantization of the energy spectrum of the electrons, and transverse deflection of the particles by the laser. Other than the choice of silver as the material, two of these observations are repetitive of prior art (notably Refs. [7] and [27]). One or more of these topics might be collected into a separate paper and/or some of them moved into the supplemental results. In any case, I feel that revisions are needed to improve how these topics are motivated, how the interrelations between them are explained, and how and where they are organized and placed within the manuscript relative to each other.

2. A central point of the authors’ approach in developing structures with metallic components is to take advantage of plasmonic resonant enhancement. However there is relatively little description of this effect or how it is displayed in the structures studied. Similarly the use of silver as the metal of choice is only briefly discussed and seems poorly motivated. Similarly there is little explanation of how the structures are fabricated. Even if only mentioned briefly in the text, perhaps some supplemental material could be included on this.

3. There are a lot of figures presented, and not all of them appear critical to the main points of the paper. A number of these (S1 through S6) have been collected at the end of the manuscript following the references. This suggests that these are “supplemental” figures provided for interested readers to gain more information. However, there are so many references to these figures within the text that they must be repeatedly examined to understand what is being discussed. In fact the first figure that is referenced in the paper is one of these supplemental figures, which is quite an odd experience for the reader. I recommend a re-evaluation and possibly a culling of the figures and a re-organization of them to include within the body of the paper all figures that are essential for understanding of the results and to substantially reduce explicit reference to figures that are not within the main text.

4. There is repeated reference to Figure 1(f) which does not exist. It appears that the intended reference is Fig. 2(a).

5. In all plots of electron energy spectrum, the horizontal axis displays the initial energy subtracted from final energy (or negative energy change). This convention is common in EELS and electron microscopy but may be unfamiliar to others. I would suggest to mention this choice of convention in the text early on and perhaps to add some feature to the first such plot explicitly indicating that energy “gain” is to the left and energy “loss” is to the right.

6. The text makes numerous references to spatial dimensions using terms such as “distance-dependent” or taking “space” as an axis label on Fig. 2(b) and “distance” in Fig. 3. The authors have defined a Cardinal x, y, z coordinate system. I recommend to use this to replace or augment many of these verbal descriptions with reference to specific coordinates.

7. In Figure 3 why are the quantized energy peaks visible in 3(c) but not in 3(b)? It is stated that the difference between these plots is in the fluence of the incident laser. But it appears there is also a substantial difference in overall amplitude (or integrated area) of the spectral distribution. Otherwise put, why do the curves at the tops of both plots not look more similar, since regardless of the laser fluence applied, in the limit where the electron beam is far from the surface the two results should in principle converge to the same (zero loss point) spectrum.

8. In relation to Fig. 3(a) the authors say in the text that “the microstructural features of the surface plasmon fields clearly show the occurrence of energy- and distance-dependent alterations along the surface normal direction.” What are the “microstructural features” of the surface plasmon fields and how are they purported to relate to what is observed in this figure?

9. For the inelastic scattering results depicted in Figure 4, why are the undeflected electron spots so different between the two cases (i.e. comparing size of the beam spot in top-most plots of (a) and (b)

panels of the figure)? Similarly for this section, the meaning of the figure in Part (c) is not well explained. Also it is not clear why the top and middle plots of Figure 4 are subtracted to form the bottom plots or what additional understanding is provided by these “Difference” plots.

10. Still regarding the results from Figure 4, it is stated in the text that the laser is at an “oblique” angle. However, in the insets on Figure 4(a) and 4(b) the laser appears to be at normal incidence or slightly off of normal. Usually when “oblique” is used in this context it means oblique relative to the surface. I recommend to clarify this. I also recommend to add a larger figure of the experimental arrangement.

11. The authors allude to a theoretical model for the observed “x-shaped” angular distribution of the electrons in the transverse coordinate space. However the explanation is unclear. For example, it is stated that the “angle between two scattered electron beams is approximately equal to the angle between the incident and reflected laser beams.” What does this mean, given that there is only one electron beam (not two) and if the laser beam is “reflected” then wouldn’t it be twice the angle θ in the figure and shouldn’t it also be diffracted into various diffraction lobes upon reflection?

12. Which MLA structure is used in the experiment of Figure 4?

13. For the bowtie MLA experiment, two different gradient values are given in the abstract and in the text. The abstract and Table S1 claim a gradient of 0.333 GV/m. However on line 248 it is stated that the gradient is 0.06 GV/m (which is substantially lower) whereas the MLA gradient “could reach 0.335 GV/m”. Is the 0.335 GV/m number a theoretical extrapolation? The authors need to make it clear what the experimental result of this paper is.

14. As the terms “beam” and “spectrum” apply to both the electrons and photons it would be advisable to make clear distinction when using these and related terms as to which is being referenced when it is not clear from context.

15. Figures 5(c) and (d) are arranged so as to invite a direct comparison between them, with one being data and the other being a simulation/calculation. However, the inset figures within these two panels display plots that represent significantly different scenarios. The data are shown for an excitation closer to the MLA surface and over a wide energy range, whereas the corresponding simulation plot shows the quantized energy states observed at very low laser fluence. This is confusing and it isn’t clear what the reader is expected to understand by comparing these two plots.

16. While the explanation for the observed asymmetry in the spectrum of Figure 5 in relation to nonlinear second harmonic (SH) generation is interesting and convincing, the paragraph explaining this effect

(beginning on line 262) talks about 2-color excitation, which confuses the reader and sends him searching back through the paper to understand where a second color of excitation comes from, since it was not previously mentioned or included in any diagram. The authors then conclude the paragraph by explaining the connection to the observation via SH generation. The ordering of this seems to me inverted. Why not tell the reader immediately what the explanation is and then justify it with subsequent comparison with two-color excitation. Similarly, various explanations within the paper tend to follow a similar trend, where details about seemingly unrelated topics are presented up front only to tie them to the topics of the paper at the end.

On the whole, I feel that the authors have done quite a bit of good work and there are some clearly interesting new results in the paper. However I think substantial revisions of the manuscript and figures are needed to make it clear and understandable to a broad audience.

Response to Reviewers' Comments

Reviewer #1 (Remarks to the Author):

The authors present an experimental study of a plasmonic metasurface laser accelerator. They show experimental studies involving two types of MLAs – “conventional” gratings and a periodic array of bowtie antennas, exhibiting acceleration gradient of up to 180MeV/m and 335MeV/m respectively. In additions, the authors discuss the possibility to excite the MLA with a field comprising a superposition of two harmonics, showing that such approach can lead to asymmetric features in the EELS spectrum which can be controlled by the phase shift between the two frequencies.

As a general comment, I believe that this is an important work. To my knowledge this is the first experimental demonstration of an MLA. Furthermore, that fact that this first experiment exhibits normalized acceleration gradient (or “efficiency” as the authors call it) which is in the same range as the best DLA result, indicate that the MLA approach could be advantageous for applications where high pump power are inapplicable. In that context, I believe the manuscript should be published pending the modifications and the revisions detailed below.

Answer: Thanks for the positive comments.

Most of the manuscript is clearly written and well explained. However, there are a few less-clear sections that should be improved for the benefit of the reader. Specifically, the section dealing with the inelastic scattering of free electrons by an incident laser is not clear. Figure 4c is also puzzling – I found it quite difficult to understand the authors meaning and what the process presented in this figure is.

Answer: Fig. 4c is schematically present the longitudinal and transverse momentum variation of electron after interact with MLA. In order to focus the investigations on main issues in present paper, we have deleted the Fig. 4 and rewritten the related discussion in revised version.

There are several missing pieces of information which must be included:

1. In Fig. 1 the authors show the calculated field in the direction of the acceleration. The amplitude of the field in the orthogonal direction (perpendicular to the MLA) should also be presented and discussed. Such field induces deflection forces that can modify the particle height above the surface and, hence, the average acceleration gradient.

Answer: The three components of field, ε_z , ε_x and B_y are shown in Fig. S3 (Page 19), as also illustrated in following figures. In Fig. 1e, the deflection forces don't count in our calculations, and the electron beam is set to be ~ 20 nm (impact factor = 20nm) from the surface.

2. The manuscript lacks a comprehensive theoretical analysis of the expected energy gain/loss spectrum. Specifically, it is not clear whether the theoretical analysis in Fig. 5 is based on a complete particle tracking simulation, 1D relativistic “Newton-law” based calculation, etc. It is also unclear whether the theoretical analysis assumes a specific height of the electron beam above the MLA or does it take into consideration the distribution of the electrons in the E-beam. Are the perpendicular deflection forces

considered in the model? Do the authors consider particles that hit the metallic structure or electrons which move away from the MLA and do not reach the detector?

Answer: Theoretical analysis in Fig. 1 and 2 is based on the conventional theory, i.e. 1D relativistic “Newton-law”, as shown in SI 1. Theoretical analysis in the Fig. 5 (Fig. 4 in revised version) is obtained using the multimodal PINEM theory (SI 2), which does not represent the particle tracking views, as reported by Konečná, A., et al. [*ACS Photonics* 2020, 7, 1290]. Though we didn’t directly consider the distribution of the electrons in the beam and the perpendicular deflection forces, we assumed that electrons interact with near-fields at different coupling parameters g as illustrated in Fig. S5a and SI 2.3. Therefore, the mentioned effects are partially considered in our theoretical discussion. In our case, the electrons that hit the metallic structure which cannot reach the detector have been not considered.

3. Some technical details on the laser and electron beams are missing. What is the size of the E-beam? Where is it aimed at with respect to the MLA (height above the surface)? What is the amplitude of the driving field (this piece of information is hiding in the SI – it should be clearly indicated in the main text)?

Answer: The electron-beam is focus to an area with a size of ~ 45 nm (FWHM) and it is about 22 nm from the surface. The amplitude of the driving field is 0.978 GV/m (spot diameter: 50 μ m (FWHM), fluence: 25.5 mJ/cm², the repetition rate: 200 kHz, the temporal width: 200 fs (FWHM), we can get an intensity of 1.27×10^{11} W/cm². We add an illustration in the main text (page 4):

In our measurements, the main experimental parameters are as follows: the initial electron beam is always centered at 200 keV with width of ~ 2.4 eV, focus to a diameter of ~ 45 nm (FWHM) and at about 20 nm (i.e., impact factor) from the MLA surface for EELS requirements; the electron pulse has a temporal width of 350 fs (FWHM) as measured by PINEM²⁵. The pump laser is always centered at the wavelength of 515 nm with a repetition rate of 200 kHz, a spot diameter of ~ 50 μ m (FWHM), a temporal width of 200 fs and a fluence of 25.5 mJ/cm². Therefore, the laser intensity is 1.27×10^{11} W/cm² and driving electric field is 0.978 GV/m.

4. There is an apparent “discrepancy” in the choice of the grating period. Assuming $\beta=0.7$ and $\lambda=515\text{nm}$ the periodicity should be 361nm and not 358nm . To resolve this, one needs to calculate the precise β of the electrons (0.695 for 200keV) to find that the period should be 357.8nm – corresponding to the choice of 358nm of the authors. I suggest the authors use the precise numbers to avoid confusion.

Answer: Thanks for your recommendation. In our study: $\beta = 0.695$, not 0.7 .

5. Referring to Fig. 5c, the zoom-in inset does not show the photon energy quantization which is expected according to inset of Fig. 5d. The authors should explain why they are unable to show this, particularly in view of the high-resolution EELS their apparatus exhibits, as seen in Fig. 2a.

Answer: We have revised Fig. 5 (i.e., Fig. 4 in the revised version) for the sake of comparison and discussion. The inset figure in Fig. 5c (Fig. 4d in revised version) was obtained by the electron beam very close to MLA surface (impact factor ~ 0), which is not comparable with the insert of Fig. 5d (Fig. 4f in revised version).

1. The visibility of the separated peaks in asymmetric spectrum depends on spectrometer resolution setting for experimental data requirements. For instance, if we use strong laser power, the spectrum would be reaching several keV, we often use larger spectrometer dispersion to collect electrons in large energy scale, this operation would submerge the quantum peaks in spectrum [Adiv, Y. *et al. PhysRevX*. 2021, 11, 041042].

In fact, we can obtain the results with quantum peaks with a resolution setting measurement, as shown in the lower panel of Fig. 2a.

2. If we lower the laser power, certain features became invisible for the second harmonic generation.

3. It is worth pointing out that we can also obtain the asymmetry separated peaks on the spectrum of a silver nanowire [Zheng, D. *et al. Nano Letters* 2021, 21, 10238].

6. Referring the same figure, it is not clear what the fundamental and second harmonic are. Is it 515nm and 258nm or 1030nm and 515nm? Furthermore, what are the intensities of the two beams?

Answer: The fundamental is 515 nm and second harmonic is 257 nm. The incident laser is 515 nm with an intensity of 1.27×10^{11} W/cm², other parameters of incident laser are shown in page 4. The second harmonic field is generated by the nonlinear optical effect in this bowtie structure, its intensity can be obtained from the theoretical simulation. Because the electron pulse duration is longer than laser pulse, electrons actually interact with the nonuniform near field. We assume the probabilities of electron in different near-field could be expressed as a linear relationship, the coupling parameter $|g_1|$ ranges from 0 to 700 in our simulation (see Fig. S5a), thus, weighted average $|g_1|$ equals to 233 (corresponding $|G_1|=0.156$ GV/m, according the equation (2) in main text of revised version). The correlation of fundamental field $\mathcal{E}^{(1)}$ and SH field $\mathcal{E}^{(2)}$ could be expressed as $|\mathcal{E}^{(2)}|=k \cdot |\mathcal{E}^{(1)}|^2$, k is the fitting coefficient and $k=0.78$ m/GV in our simulation, we can get $|g_2|=23.3$ (corresponding $|G_2|=0.0312$ GV/m), and $|G_2|/|G_1|=1/5$. As a result, it is in good agreement with experimental results in Fig. 4c. So, we inferred that the intensity of SH near field is 1/25 of fundamental field.

7. With respect to comments 5 and 6, the claim at the end of the section dealing with Bowtie structure MLA – “They showed that the existence of the SH field would induce transitions into specific states, resulting in an asymmetric energy state distribution”, is not well-supported by the experimental results.

Answer: It is a result of theoretical simulation in the inset of Fig. 5d (Fig. 4f in revised version), the related experimental results didn't been shown here.

8. There is a typo referring to Fig. 1f (lines 134, 135, 141). I assume the meaning is Fig. 2a.

Finally, one of the great promises of MLA is their alleged ability to provide “efficiency” that exceeds unity (i.e. obtaining normalized acceleration gradient which is larger than

1). Although the results presented by the authors are impressive, they still fall short compared to those of contemporary DLA structures. I expect the authors to provide a detailed discussion of the mechanisms limiting the acceleration gradient of their structures and what can be done in order to improve it. Does the reason for the limited gradient stem from the fact that the MLA is not tapered? Does it have to do with the choice of material (Ag)? The unit-cell structure (bowties and not, say slit antennas as has been suggested in previous theoretical studies)? etc.

Answer: Thanks for your reminder. It had been revised.

In theory, the MLA indeed has the ability to provide normalized acceleration gradient larger than 1, due to the local field confinement and enhancement. In experiment, several factors must be considered: 1. The MLA nanostructure must be optimized to get high electromagnetic fields coupling with electrons 2. Nanofabrication precision. 3. Quality of electron beam. In order to take advantage of this strong local near-field in the nanostructure, electron beam must to focus in a diameter less than 10 nm with a small divergence angle. For long distance acceleration, accumulated phase-mismatching effect must be also considered. Actually, silver, gold, aluminum, and other metallic material, they all have the high reflection rates, high conductivity and low absorption rate at certain wavelengths, so they could also be used for accelerating electron. The unit-cell structure can be different from bowties, and theoretical simulation should be carried out for the specific MLA nanostructure to get better properties.

Reviewer #2 (Remarks to the Author):

The authors describe their experiments of facilitating silver-based nanophotonic structures (antennas) to manipulate 200 keV electrons in an ultrafast transmission electron microscope, with the stated goal of having these structures (and in particular: metal) as an electron accelerator on-chip. Metals have many advantages over dielectrics for serving as electron accelerators, such as high conductivity and the ability to create smaller features (when discussing MLAs, as is the case of this work). However, there is one great disadvantage, which is the much lower damage threshold of metals versus dielectrics.

This manuscript presents well the authors' experiments and is an important contribution to the on-chip nanophotonic electron accelerator community. However, there are two fundamental difficulties that must be addressed prior to publication (which I fully support! Great results!). These are namely: the damage threshold and nonlinear effects.

Answer: Thanks for the positive comments.

Below I list minor comments to be addressed by the authors, and conclude with the above main questions.

First, throughout the manuscript, the authors should differentiate between acceleration (GeV/m) and field strength (GV/m).

Answer: It had been revised, and the unit for electron acceleration (GeV/m) is used.

Abstract:

Mentions the acceleration gradient of 0.335 GeV/m, but should also reflect in a sentence the reason for this (maximum) gradient. What is the damage threshold?

Answer: The damage threshold ranges from 1.1 to 1.6 GV/m under the laser radiation with a wavelength of 515 nm. It depends on the fabrication. We revised the statement in abstract (page1):

Due to the presences of the surface plasmon enhancement and nonlinear optical effects, the maximum acceleration gradient reaches to 0.335 GeV/m.

Page 3:

- The authors write,

“However, the required acceleration gradient on the gigavolt per metre scale has not been realized by DLA to date, and the exploration of new laser-driven accelerator designs continues.”

In Cesar, Comm. Phys. 1 (2018) (<https://doi.org/10.1038/s42005-018-0047-y>) the authors demonstrated up to 9 GV/m driving fields with 1.8 GV/m in the accelerating mode. Due to technicalities in the design and experiment, the average gradient is 0.85 GeV/m. Although strictly speaking the average gradient is below 1 GeV/m, I would suggest to rephrase the above sentence to reflect the results of Cesar. In general, the authors and readership may find the recent DLA review paper in Advances in Optics and Photonics [<https://doi.org/10.1364/AOP.461142>] useful, as it contains a table of recent DLA experimental works.

Answer: The relevant statements have been revised as follows:

Although the driving fields have reached 9 GV/m, the average acceleration gradient at the scale of GeV/m has not been realized by DLA to date^{16,17}, and the exploration of new laser-driven accelerator designs continues.

- Still on page 3, “It should also be noted that the threshold gap between metallic and dielectric materials decreases” – what is a threshold gap? This should be corrected/the definition should be added to the manuscript.

Answer: Here threshold gap means the difference of damage thresholds between metallic and dielectric materials. It is revised in page 3:

It should be also noted that the difference of laser damage thresholds between metallic and dielectric materials can be lowered with the decrease of temporal widths of pulsed laser^{4,5,18}.

Page 4:

- The authors mention an energy gain of 150 eV when interacting with a single silver nanowire. The laser wavelength (photon energy) and electron energy should be mentioned alongside this.

- Also, the authors make a sharp jump from the first few sentences about the silver nanowire to “Figure 1a shows. MLA”. I would suggest to make a smoother transformation by adding one more sentence before the description of Figure 1.

The details of the laser (pulse length, wavelength, photon energy, spot size on sample, intensity) should be mentioned here. One could start this section by a short description of all the experiments to be detailed in it.

Answer: The laser wavelength always is 515 nm (2.4 eV) and electron energy always is 200 keV. A sentence has added in the caption of Fig. S1 as follow:

The Electron and laser beam parameters are the same with the parameters demonstrated in the main text.

We also add a statement in page 4:

previously, we have performed an investigation on a silver nanowire, which can be approximated as an MLA with only one period. It was observed that the electron energy-gain for a single silver nanowire with a diameter of approximately 100 nm can be as high as 150 eV or more, as shown in Fig. S1; this illustrates the great potential for metallic nanostructures used in development of laser-driven accelerators.

The details of the fs- laser are shown in Page 4.

Page 5:

- Figure 1c – add a scale bar.

- “but this type of standing wave cannot be attributed to electron acceleration (see the Supplementary Information (SI))” – please provide the correct section in the SI. This comment should be taken into account anywhere in the manuscript where a reference to the SI is made.

Answer: A scale bar is added.

The revised statements of SI have been added in all relevant references.

Page 6:

- “When the electron movement direction is opposite to the optical field direction near the MLA surface” – this is confusing: the electron trajectory is always towards +z. Do you mean to say, “when the polarity of the optical field is negative”? Should be rephrased.

- “In fact, periodic acceleration and deceleration of this type would compress a femtosecond-width electron pulse into a series of attosecond pulses²⁸” – actually, free-space propagation is required for this type of ballistic bunching, as shown in Ref. 28.

- “This formula can explain the experimental results in the upper panel of Fig. 1f well” – there is no panel (f) in Figure 1?

- The authors define the relationship between g and G as dependent on the number of periods N (also derived in the SI). This has a hidden assumption, that there is no dephasing (in this periodic case - electron energy doesn't change during interaction). I suggest to add this remark, especially when making the calculation to $g=833$.

Answer: The related statements have been revised as follows:

When the z-component of optical field nearby the MLA surface goes along the reverse direction of the z-axis, electron 1 can be accelerated.

In fact, the existence of periodic acceleration and deceleration in present case could compress a femtosecond-width electron pulse into a series of attosecond pulses after a distance of free-space propagation.

It is Fig. 2a, not Fig. 1f.

Ignoring the dephasing from the electron energy variation during interaction, the relationship of g to G can be written as:

Page 7:

- Figure 2: it is not clear what “space” means in panel b. It is also imperative that the authors add a comparable measurement with laser off, both for panels (a) and (b), to stress the differences. What is the color scale in panel (b)?

Answer: Here “space” means distance along the x-axis direction. A spectrum with laser off is added in Fig. 2a. A laser-off image only shows up a narrow peak along the energy dimension, as shown in below. The color bar is the log scale, data are normalized.

Page 8:

- Figure 3: please use a line to mark the edge of the MLA in the panels, or alternatively add a panel with a “normal” (non-energy-filtered) image. What is the curious oscillation we see at the bottom of yellow-blue electron cloud?

- What does the y-axis “distance” in Fig.3b and c mean? Which distance?

- The authors promise to “discuss extensively” the asymmetric spectra in Fig.3b, however, I could not find such a discussion.

Answer: A red dashed line has added to marks the edge of the MLA in Fig. 3a. Here the oscillations at the bottom of yellow-blue electron are arising from micro-structure of MLA with the uneven surface.

The “Distance” means impact factor, i.e., the distance between electron spot center to edge of MLA (see Fig. 3b).

According to our observations, asymmetric spectra can be obtained in both the simple grating structure and bowtie structure, so we had added a sentence in discussion of page 10:

We attribute the asymmetric structure in Fig. 3d, Fig. 4c-d to optical nonlinear effects, i.e., the second-harmonic (SH) field with wavelength 257 nm generated on MLA induced by incident field.

Pages 8-11 and Figure 4:

- I assume that at $\Delta t = -6.7$ ps the laser light and electrons do not interact (at all).
Why are Fig.4a and Fig.4b (first row) not identical?

- Please add a line to depict the interface of the MLA in the panels.

- The explanation of Figure 4 is not very clear. It seems to me that when the laser is incident at an angle to the MLA, the incident and reflected beams generate transverse components of the electric field, which can explain the “X” shape. Is this kind of classical description correct?

- Similarly, the deflection (scattering) in the direction normal to the MLA surface (we see this particularly in the left column): could this be a result of the forces acting on electrons that are not at the peak of the electric field, and are so deflected “into” or “away” from the MLA surface?

Answer: In our experiments, the pulse duration of the electron and laser is hundreds of femtoseconds, so no interaction between electron and near fields appears at $t = -6.7$ ps.

The difference is that the undeflected electron spot in Fig. 4a is larger than the undeflected electron spot in Fig. 4b

As stated in above context, we have deleted Fig. 4 in the revised version. We are carrying out another work which will focus on the momentum transfers between optical fields and electrons in MLA.

Pages 11-14 and Figure 5:

- Page 11: “In our recent study” – did you mean, “in this work”? Otherwise, supply a reference.

Answer: It has been deleted.

This is the core result of this work: showing a nanophotonic antenna interact with an electron. The authors have two claims here:

(1) Nonlinear effects distort the spectrum towards energy-gain.

I understand from the authors that they assume their “fundamental” on the sample (i.e. 515nm light) goes through a nonlinear process and generates its second-harmonic (257nm). These two interact and are responsible for the energy-spectrum distortion of

the electrons. I see some issues with this claim, which to me makes this claim tough to accept:

What is the efficiency of the SH-generation? What would be the calculated field amplitudes (in GV/m) for each?

Answer: The efficiency of SHG in this bowtie structure depends mainly on the intensity of laser. In our theoretical simulation, the weighted average fundamental acceleration field is 0.156 GV/m (corresponding $|g_1|=233$), SH acceleration field is 0.0312 GV/m (corresponding $|g_2|=23.3$). Therefore, the SH near field is one-fifth of fundamental field. Also see the answer for question 6 of reviewer#1.

What is the decay length from the antenna for 515nm, as opposed to the decay length for 257nm?

Answer: Theoretically, the decay length can be estimated by $\delta = \frac{\beta\gamma}{k_0}$, we can get the decay lengths of 79.2 nm for 515 nm laser and 39.6 nm for 257 nm laser.

How does the doubled wavelength (257nm) impact the phase-matching condition with the electrons?

Answer: Phase-matching condition for this grating structure, $\Lambda/n = \beta\lambda$, where Λ is the spatial period, λ is the light wavelength, n represents nth spatial harmonic oscillates of grating. In our case, $n = 1$, thus, $\Lambda = \beta\lambda$ is tenable. The phase-matching condition of SH field is $\Lambda/2 = \beta\lambda/2$, i.e., SH field matches the 2th spatial harmonic oscillation ($n = 2$).

What is the efficiency of coupling of the fundamental (515nm) versus the SH (257nm)?

Answer: The relationship of fundamental field $\mathcal{E}^{(1)}$ and SH field $\mathcal{E}^{(2)}$ could be written as $|\mathcal{E}^{(2)}|=k \cdot |\mathcal{E}^{(1)}|^2$, k is fitting coefficient. According to the fitting result, $k=0.78$ m/GV. Also see the answer in question 6 for reviewer#1.

What is the absorption in Silver for these two wavelengths?

Answer: For the normal incident 515 nm light, the absorption coefficient of Silver is 0.078 nm^{-1} , the reflectance is 0.98 [Phys. Rev. B 91, 235137 (2015)]; For 257 nm laser, absorption coefficient of silver is 0.068 nm^{-1} , the reflectance is 0.28 [Phys. Rev. B 6, 4370-4379 (1972)].

Why is it true that the energy distortion is towards the energy gain, rather than loss?

Answer: If the relative phase of the fundamental and SH fields δ is zero, Spectral enhancement is towards the energy gain [Konečná, A., et al. ACS Photonics 2020, 7, 1290].

Why isn't it a symmetric distortion? How can you experimentally control whether the distortion is towards this or that direction?

Answer: The probability of energy electrons in each state depends on the intensity of fundamental near-fields, the SH generation and the relative phase δ . If $\delta = \pi/2$, we can obtain a symmetric structure; if $\delta = \pi$, the energy distortion would toward energy-loss (see Fig. S4). Experimentally, the phase of SHG depends also on the materials and grating structure.

Were Figure 5d and Figure S6 based on theoretical calculations or a full numerical COMSOL simulation, including the SH generation?

Answer: Fig. 5d (Fig. 4e and f in the revised version) and Fig. S6 (Fig. S4 in revised version) are the theoretical calculations with the SH generation.

I see the authors' claim with Figure S6 (which is a simulation, if I'm not mistaken), and the calculations, but can you prove (at least by answering the questions above) that this is indeed the case in the experiment? It might be that you can indeed get this result from a combination of 515nm+257nm, but I am not convinced yet that this is the case. I suggest to look at particle-tracking simulations (please include in the SI) and see if e.g. more of the decelerating particles "crash" into the structure, leaving a higher number of accelerated particles to reach the energy analyzer. The angle of divergence might also

affect the energy analyzer in experiment, by misinterpreting high angles with energy shifts (what aperture did you use?) or missing the energy analyzer altogether.

Answer: Yes, Figure S6 (S4 in revised version) is a result from a combination of fundamental field + SH field. In our experimental measurements, no apertures were used, the electron beam collection angle is larger than 100 m rad.

We have confirmed our data used single nano-wire in which no electrons can “crash” in the samples to impact the spectroscopic analysis [*Nano Letters* 2021, 21, 10238].

(2) Damage threshold of Silver in the experiment.

The technical details the authors provide did not convince me that their structures can withstand 1 GV/m incident fields. As a result, I am not convinced that the acceleration is indeed 335 MeV/m.

On page 11, the authors write:

“The MLA acceleration gradient could reach 0.335 GV/m, i.e., approximately 1/3 of the incident field. This indicates that the prototype MLA can generate an acceleration gradient and efficiency comparable with that of the highest performance DLA^{6,9,43,44.}” First, note again the work by Cesar with 9 GV/m fields. This claim should be changed. How did you calculate “approximately 1/3 of the incident field”? Please convince me by adding a section to the manuscript (or SI if you must, but the expert readers would appreciate it in the manuscript) with a complete detailed list of your laser parameters (average power, beam size in x,y, repetition rate, pulse length) on the sample (i.e.: your second-harmonic = 515nm), and the formula you used to convert this to the peak optical field in GV/m. Also mention the peak intensity and fluence. Please also mention the electron pulse length (FWHM) and beam size (FWHM) and how you measured each.

Answer: We have revised this part as:

The MLA acceleration gradient could reach 0.335 GeV/m, i.e., approximately 1/3 of the incident field. So far, this value is highest record of average acceleration gradient in the single grating structure-based laser-driven accelerator¹⁶, this fact suggests that the prototype MLA could have high performance in the developments of compact accelerators.

Details of the electron beam and laser parameters is illustrated in the main text (page 4). Theoretical analysis for optical fields is shown in following context:

In experiment, we can measure laser power at a position close to laser port of UTEM chamber. The relation of laser peak intensity with laser power can be written as $I =$

$\frac{P}{r_{\text{repetition}} \cdot T_{\text{duration}} \cdot \pi(d/2)^2}$, where P is the measured laser power, $r_{\text{repetition}}$ is repetition

rate of laser, T_{duration} is laser pulse duration, d is laser spot diameter. In other hand,

peak intensity could also be written as $I = \frac{1}{2} \epsilon c n E^2$, where ϵ is dielectric constant, c

is light speed, n is refractive index. Therefore, the peak optical field is $E = \sqrt{2I/\epsilon c n}$.

Taking the parameters $P=100\text{mW}$, $r_{\text{repetition}}=200\text{kHz}$, $T_{\text{duration}}=200\text{fs}$, $d=50\mu\text{m}$,

$\epsilon=\epsilon_0$, $n=1$, we have $I=1.27 \times 10^{11} \text{ W/cm}^2$, and $E=0.978 \text{ GV/m}$.

The electron pulse (FWHM) of $\sim 350 \text{ fs}$ is measured by PINEM method, as shown in below left; electron beam size with diameter $\sim 45 \text{ nm}$ is directly measured by camera, as shown in the following image (right panel).

Photonic nanostructures have a different damage threshold than flat surfaces. The authors have surely burned more than one of their structures. It would only strengthen the authors' claim if they add a section to the SI showing several images of the structures after laser irradiation, comparing before- and after-damage and showing the types of damage they see, and at which laser parameters.

Answer: Yes, it is known that the photonic nanostructures have a lower damage threshold than flat surface. Several images before and after laser irradiation shown in in Fig. S2 in revised version. The fluence is 50.9 mJ/cm^2 (the corresponding intensity is $2.55 \times 10^{11} \text{ W/cm}^2$, driving electric field is 1.38 GV/m). A sentence is added in page 5 as follow:

A comparison of MLA before and after laser damage is shown in Fig. S2.

How did the authors deduce an acceleration to 2.4 keV? It's most important to be very careful here. Figure 5c shows this, but I would like to see a close-up on that acceleration tail. What is the noise level of this measurement? Can you assert that you are above the noise level and, that this feature is indeed a “real” measurement and not an effect of aberrations through the spectrometer (e.g. high-angle spatial aberrations translate to energy)? A corresponding measurement of the spectrum super-imposed on this curve, with laser off but otherwise exact same parameters, would help quite a lot here.

Answer: 1. A close-up of acceleration tail is shown in below and Fig. 4c in the revised manuscript. It is shown that clear signal can be seen at 2.4 keV above the noise level. Noise level of this measurement is 31 counts (standard deviation).

2. The highest angle the transverse deflection angle can reach to 4.8 milliradians (absorbing 1000 photons). It is calculated by $\theta \approx \arctan \frac{n\hbar k}{p_0}$, where p_0 is initial momentum of electron, $\hbar k$ is the momentum of photon, n is the photon number. For conventional EELS experiments, the zero-loss peak contains electrons with scattering angle from zero to more than one hundred milliradians [Egerton R 1996 *Electron Energy Loss Spectroscopy in the Electron Microscope*]. This fact indicates that high scattering angle electrons can be also correctly characterized by EELS. In addition, electrons with several keV can be well characterized by Gatan imaging filter [<https://live-eels.pantheon.io/products/gatan-imaging-filter-gif>].

3. A relevant measurement of the spectrum at $t = -6.7$ picosecond delay is shown in following figures, which is similar with the laser-off result in Fig. 2a in the revised manuscripts.

In the conclusion, the authors write “A prototype of the proposed MLA provides a new way to realize compact accelerators on chip.” I agree: MLAs have great advantages over dielectrics. But, the authors should still remind the reader of the damage threshold: a conclusion from this work is still, that dielectric materials have been experimentally shown to endure at least 1 order of magnitude larger laser fields.

Answer: At the laser wavelength of 515 nm, it is noted that the dielectric materials don't always show a visibly larger damage threshold than metallic materials. According our experimental results. We added a statement in page 12:

A prototype of the proposed MLA provides a new way to realize compact accelerators on chip, but it should be noted that dielectric materials generally have higher laser damage threshold.

Reviewer #3 (Remarks to the Author):

In this paper, the authors present experimental demonstration of electron acceleration in the vacuum region above a microstructured device excited by an optical laser pulse. Two structures are considered: one is a simple metal grating and the other is a sequence of “bow-tie” structures arranged in a periodic 2D array on a planar substrate. The authors also make some interesting observations about transverse deflection of the particles under interaction with the laser in vacuum. In recent years, a variety of related experiments have been conducted by various researchers, primarily using structures made of silicon or silicon dioxide. Metals have been generally avoided due to damage limits of the laser. The authors here point out that some metals can have damage thresholds comparable to silicon under certain laser parameter conditions and thus can provide similar levels of “accelerating gradient” to a charged particle. In addition a plasmonic enhancement effect can potentially allow for more efficient excitation of the accelerating field. The authors include a comparison of their results against some of this prior art. While some aspects of the work appear repetitive of experiments conducted by previous researchers, the most novel and interesting aspect presented in this paper is the use of the metallic bowtie structure. The novel aspects merit publication, and the general treatment of the subject matter seems appropriate for Nature Communications. However, the structure and wording of the paper is often confusing and I believe requires substantial improvement in organization, choice and placement of figures, and the manner in which the results are explained and motivated.

Answer: Thanks for the positive commends.

I outline below various points that I feel should be addressed by the authors.

1. The abstract emphasizes the so-called bowtie structure as the primary result. However, the authors spend a substantial part of the paper initially describing various other results, including interaction with a nanowire and with a metallic grating, quantization of the energy spectrum of the electrons, and transverse deflection of the

particles by the laser. Other than the choice of silver as the material, two of these observations are repetitive of prior art (notably Refs. [7] and [27]). One or more of these topics might be collected into a separate paper and/or some of them moved into the supplemental results. In any case, I feel that revisions are needed to improve how these topics are motivated, how the interrelations between them are explained, and how and where they are organized and placed within the manuscript relative to each other.

Answer: In order to focus on the acceleration results, we have deleted Fig. 4 in revised manuscript and omitted the relevant qualitative analysis on the longitudinal momentum changes. For better understanding of nonlinear optical effect in MLA and highlight the asymmetry electron spectra, we have revised the discussions and added figures about bowtie structure MLA.

2. A central point of the authors' approach in developing structures with metallic components is to take advantage of plasmonic resonant enhancement. However there is relatively little description of this effect or how it is displayed in the structures studied. Similarly the use of silver as the metal of choice is only briefly discussed and seems poorly motivated. Similarly there is little explanation of how the structures are fabricated. Even if only mentioned briefly in the text, perhaps some supplemental material could be included on this.

Answer: Previously, some theoretical works have demonstrated that metallic nanostructure can be applied in laser-driven accelerator (see the reference 21-23 of manuscript). MLA takes advantage of this effect using phase-matching condition, i.e., the spatial period = $\beta\lambda$.

In paragraph 2, we have discussed the main ideas for using metallic materials. The silver as an e-p interaction media is motivated by our previously works on the Ag-nanowire [*Nano Letters* 2021, 21, 10238]. It is shown that a single silver nanowire could withstand intense laser radiation and accelerate electron to gain energy more than 100 eV, this fact has promoted our study on the laser-driven accelerator by metallic materials.

We had revised and added some sentences as follow in page 4 to 5:

Previously, we have performed an investigation on a silver nanowire, which can be approximated as an MLA with only one period. It was observed that the electron energy-gain for a single silver nanowire with a diameter of approximately 100 nm can be as high as 150 eV or more, as shown in Fig. S1; this illustrates the great potential for metallic nanostructures used in development of laser-driven accelerators. Under the laser radiation, the responding of free electrons in the metal could make the electromagnetic field redistribution, which allowed metallic material could focus and confine electromagnetic wave on some local areas and generate strong near fields. To take advantage of plasmonic field enhancement, the structural period of the MLA should be decided by the phase-matching condition: $\Lambda = \beta\lambda \dots$

The detail of structure fabrication has been demonstrated in Method part of revised manuscript in page 12.

3. There are a lot of figures presented, and not all of them appear critical to the main points of the paper. A number of these (S1 through S6) have been collected at the end of the manuscript following the references. This suggests that these are “supplemental” figures provided for interested readers to gain more information. However, there are so many references to these figures within the text that they must be repeatedly examined to understand what is being discussed. In fact the first figure that is referenced in the paper is one of these supplemental figures, which is quite an odd experience for the reader. I recommend a re-evaluation and possibly a culling of the figures and a re-organization of them to include within the body of the paper all figures that are essential for understanding of the results and to substantially reduce explicit reference to figures that are not within the main text.

Answer: Thank you for your recommendations. We had reorganized those figures, added and revised some figures in supplemental materials.

4. There is repeated reference to Figure 1(f) which does not exist. It appears that the intended reference is Fig. 2(a).

Answer: Yes, it is Fig. 2a.

5. In all plots of electron energy spectrum, the horizontal axis displays the initial energy subtracted from final energy (or negative energy change). This convention is common in EELS and electron microscopy but may be unfamiliar to others. I would suggest to mention this choice of convention in the text early on and perhaps to add some feature to the first such plot explicitly indicating that energy “gain” is to the left and energy “loss” is to the right.

Answer: Thanks for your recommendation, all energy spectra are electron energy-loss spectrum and the left of plot indicating energy gain. When we discuss the first plot, we added an illustration in page 4:

It should be noted, all spectra in this paper are the electron energy-loss spectra, in which the right part of plots is energy “loss” and left is energy “gain”.

6. The text makes numerous references to spatial dimensions using terms such as “distance-dependent” or taking “space” as an axis label on Fig. 2(b) and “distance” in Fig. 3. The authors have defined a Cardinal x, y, z coordinate system. I recommend to use this to replace or augment many of these verbal descriptions with reference to specific coordinates.

Answer: We exhibit it by using a specific dimension of Cardinal coordinate system in Fig. 2b and by “impact factor” in Fig. 3. The “impact factor” is explained in the insert figure of Fig. 3b.

7. In Figure 3 why are the quantized energy peaks visible in 3(c) but not in 3(b)? It is stated that the difference between these plots is in the fluence of the incident laser. But it appears there is also a substantial difference in overall amplitude (or integrated area) of the spectral distribution. Otherwise put, why do the curves at the tops of both plots not look more similar, since regardless of the laser fluence applied, in the limit where the electron beam is far from the surface the two results should in principle converge to the same (zero loss point) spectrum.

Answer: In order to clearly show the quantized peaks, the energy width of electron beam (zero-loss peak) should be less than one photon energy and the spectrometer should have enough energy resolution. In Fig. 3b, in order to collect all of the electron, the dispersion of spectrometer is 1 eV per channel (Fig. 3c is 0.1 eV per channel), which results in peaks merely un-separated. In Fig. 3(c), the fluence of the incident laser is rather low and the laser pulse had stretched to ~ 800 fs (larger than electron pulse width). Due to the wider pump laser pulse duration, all electrons can interact with near fields in Fig. 3c. The curves at the tops of both plots would be the same, if they obtain far enough from the surface.

In the revised manuscript, we have clearly illustrated the difference of two plots in page 9:

A series of spectra have been obtained using the same conditions as Fig. 3b, additionally, the lower laser fluence (0.2 mJ/cm^2) and wider laser pulse duration (~ 800 fs) are also used for clearly exhibiting quantized peaks. In order to clearly show the quantized peaks, the dispersion of spectrometer set to 0.1 eV per channel (1.0 eV per channel for Fig. 3b).

8. In relation to Fig. 3(a) the authors say in the text that “the microstructural features of the surface plasmon fields clearly show the occurrence of energy- and distance-dependent alterations along the surface normal direction.” What are the “microstructural features” of the surface plasmon fields and how are they purported to relate to what is observed in this figure?

Answer: Here the “microstructural features” of the surface plasmon fields refers to the qualitative decay of near field intensity within nano-scale nearby MLA surface.

The counts of energy-gain electrons show increase and gradually decay along the x-axis direction, and the high energy-gain electrons can only be observed in the areas very close to surface.

9. For the inelastic scattering results depicted in Figure 4, why are the undeflected electron spots so different between the two cases (i.e. comparing size of the beam spot

in top-most plots of (a) and (b) panels of the figure)? Similarly for this section, the meaning of the figure in Part (c) is not well explained. Also, it is not clear why the top and middle plots of Figure 4 are subtracted to form the bottom plots or what additional understanding is provided by these “Difference” plots.

Answer: The undeflected electron spot in Fig. 4a is larger than the undeflected electron spot in Fig. 4b.

In order to focus on the acceleration results, we removed fig. 4 and the part of electron transverse deflection in the revised manuscript.

10. Still regarding the results from Figure 4, it is stated in the text that the laser is at an “oblique” angle. However, in the insets on Figure 4(a) and 4(b) the laser appears to be at normal incidence or slightly off of normal. Usually when “oblique” is used in this context it means oblique relative to the surface. I recommend to clarify this. I also recommend to add a larger figure of the experimental arrangement.

Answer: Thanks for your reminder. We removed fig. 4 and the part of electron transverse deflection in the revised manuscript. A larger figure of the experimental arrangement will be shown in another article.

11. The authors allude to a theoretical model for the observed “x-shaped” angular distribution of the electrons in the transverse coordinate space. However the explanation is unclear. For example, it is stated that the “angle between two scattered electron beams is approximately equal to the angle between the incident and reflected laser beams.” What does this mean, given that there is only one electron beam (not two) and if the laser beam is “reflected” then wouldn’t it be twice the angle θ in the figure and shouldn’t it also be diffracted into various diffraction lobes upon reflection?

Answer: We use only one electron beam. However, after electrons interact with the MLA, because of transverse momentum variation of electrons, the projection of electron beam on x-y plane is clearly “separated” into two parts, one goes along the incident laser beam direction, another one goes along the reflected laser beam direction, a shape like “X”.

The angle between the reflected laser beam with incident laser beam is twice the angle θ in the figure and $2\theta \approx \alpha$. The reflected laser beam would not be efficiently diffracted into various diffraction lobes, because the spatial period Λ is less than laser wavelength λ .

12. Which MLA structure is used in the experiment of Figure 4?

Answer: It is a simple grating structure, not a bowtie structure. We have deleted Fig. 4 in revised manuscript and omitted the relevant qualitative analysis on the momentum changes.

13. For the bowtie MLA experiment, two different gradient values are given in the abstract and in the text. The abstract and Table S1 claim a gradient of 0.333 GV/m. However on line 248 it is stated that the gradient is 0.06 GV/m (which is substantially lower) whereas the MLA gradient “could reach 0.335 GV/m”. Is the 0.335 GV/m number a theoretical extrapolation? The authors need to make it clear what the experimental result of this paper is.

Answer: In the experiment of bowtie MLA, the average acceleration gradient is 0.335 GeV/m, it is an experimental result. The value of 0.06 GeV/m is result from the asymmetry of spectrum. This value is zero for conventional spectrum. The related statement has been revised as follow:

The average energy-gain of asymmetric electron spectrum reaches 0.43 keV (0.55 keV for Fig. 4d), it is zero for conventional spectra due to its symmetry.

14. As the terms “beam” and “spectrum” apply to both the electrons and photons it would be advisable to make clear distinction when using these and related terms as to which is being referenced when it is not clear from context.

Answer: Thanks for your recommendation. The term “spectrum” only reference to electron spectrum in this article. We have added “electron” or “laser” ahead the terms “beam” and “spectrum” in revised manuscript.

15. Figures 5(c) and (d) are arranged so as to invite a direct comparison between them, with one being data and the other being a simulation/calculation. However, the inset figures within these two panels display plots that represent significantly different scenarios. The data are shown for an excitation closer to the MLA surface and over a wide energy range, whereas the corresponding simulation plot shows the quantized energy states observed at very low laser fluence. This is confusing and it isn't clear what the reader is expected to understand by comparing these two plots.

Answer: Thanks for your reminder. The simulation plot in the insert figure is shown to demonstrate how these asymmetric structures are created, not for comparison. Fig. 5 has been revised to clearly show our purpose. Also see the answer in question 5 of reviewer#1.

16. While the explanation for the observed asymmetry in the spectrum of Figure 5 in relation to nonlinear second harmonic (SH) generation is interesting and convincing, the paragraph explains in this effect (beginning on line 262) talks about 2-color excitation, which confuses the reader and sends him searching back through the paper to understand where a second color of excitation comes from, since it was not previously mentioned or included in any diagram. The authors then conclude the paragraph by explaining the connection to the observation via SH generation. The ordering of this seems to me inverted. Why not tell the reader immediately what the explanation is and then justify it with subsequent comparison with two-color excitation. Similarly, various explanations within the paper tend to follow a similar trend, where details about seemingly unrelated topics are presented up front only to tie them to the topics of the paper at the end.

Answer: Thanks for your advice. We had adjusted the order of explanation and the comparison with previously works. A paragraph is revised as follow (Page 10-11):

We attribute the asymmetric structure in Fig. 3d, Fig. 4c-d to optical nonlinear effects, i.e., the second-harmonic (SH) field with wavelength 257 nm generated on MLA induced by incident field. The occurrence of high-harmonic generation in this bowtie structure have been demonstrated by optical methods in previous works^{35,36,38}. The

spatial distribution of the SH fields in single silver nanowire after the fs-pulsed laser excitation have been directly observed in real space by PINEM³⁹. This asymmetric feature of electron spectrum is also consist with the PINEM experimental measurements of two laser beams excitation (one with frequencies ω and another with 2ω)⁴⁰.

On the whole, I feel that the authors have done quite a bit of good work and there are some clearly interesting new results in the paper. However, I think substantial revisions of the manuscript and figures are needed to make it clear and understandable to a broad audience.

Answer: Thanks for the positive commends. We have revised manuscript and relevant figures.

REVIEWER COMMENTS

Reviewer #1 (Remarks to the Author):

This reviewer provided confidential remarks to the editor recommending publication.

Reviewer #2 (Remarks to the Author):

Please see pdf attached.

REVIEW: “Efficiently accelerated free electrons by metallic laser accelerator” – By Zheng, et. al. – ROUND 2

Rebuttal #1:

Page 7:

- Figure 2: it is not clear what “space” means in panel b. It is also imperative that the authors add a comparable measurement with laser off, both for panels (a) and (b), to stress the differences. What is the color scale in panel (b)?

Answer: Here “space” means distance along the x-axis direction. A spectrum with laser off is added in Fig. 2a. A laser-off image only shows up a narrow peak along the energy dimension, as shown in below. The color bar is the log scale, data are normalized.

The authors have added a measurement of laser-off to Fig.2a, but it raises questions. Here is the new Fig.2a from the manuscript:

What is the meaning of the one-sided glitch at the 10^2 counts line, from $x=0$ towards $\sim x=0.2$?

Rebuttal #2:

This is the core result of this work: showing a nanophotonic antenna interact with an electron.

The authors have two claims here:

(1) Nonlinear effects distort the spectrum towards energy-gain.

I understand from the authors that they assume their “fundamental” on the sample (i.e. 515nm light) goes through a nonlinear process and generates its second-harmonic (257nm). These two interact and are responsible for the energy-spectrum distortion of the electrons. I see some issues with this claim, which to me makes this claim tough to accept:

What is the efficiency of the SH-generation? What would be the calculated field amplitudes (in GV/m) for each?

Answer: The efficiency of SHG in this bowtie structure depends mainly on the intensity of laser. In our theoretical simulation, the weighted average fundamental acceleration field is 0.156 GV/m (corresponding $|g_1|=233$), SH acceleration field is 0.0312 GV/m (corresponding $|g_2|=23.3$). Therefore, the SH near field is one-fifth of fundamental field. Also see the answer for question 6 of reviewer#1.

The authors state here $|g_2|/|g_1| = 0.1$, but in the caption of Figure 4 $|g_2|/|g_1| = 0.2$. In Fig.S5, the authors pick $|g_1|=300$ and $|g_2|=26$, so $|g_2|/|g_1| = 0.087$. Please explain and then correct these discrepancies.

Could it be that the authors confuse again the acceleration gradient G (units of eV/m) with field (units of V/m)? How does the “weighted average fundamental acceleration field” G_1 of 0.156 GV/m and the “SH acceleration field” G_2 of 0.0312 GV/m, relate to the ultimate average acceleration gradient result of 0.335 GeV/m?

Assuming the authors meant that the contribution to the acceleration gradient from the fundamental is $G_1=0.156$ GeV/m and from the second harmonic $G_2=0.0312$ GeV/m, how is this weighted average calculated? It seems like the contribution of the SH is almost an order of magnitude less than the fundamental – I find it difficult to agree that the SH has such a large impact on the electron spectrum. Was Fig.4e calculated with any impact factor? Or is it a 1D simulation? This should be noted in the text.

Rebuttal #3:

What is the decay length from the antenna for 515nm, as opposed to the decay length for 257nm?

Answer: Theoretically, the decay length can be estimated by $\delta = \frac{\beta\gamma}{k_0}$, we can get the decay lengths of 79.2 nm for 515 nm laser and 39.6 nm for 257 nm laser.

How does the doubled wavelength (257nm) impact the phase-matching condition with the electrons?

Answer: Phase-matching condition for this grating structure, $\Lambda/n = \beta\lambda$, where Λ is the spatial period, λ is the light wavelength, n represents nth spatial harmonic oscillates of grating. In our case, $n = 1$, thus, $\Lambda = \beta\lambda$ is tenable. The phase-matching condition of SH field is $\Lambda/2 = \beta\lambda/2$, i.e., SH field matches the 2th spatial harmonic oscillation ($n = 2$).

Continuing the point about G1 and G2 from the previous point, now the decay lengths are half of each other, plus the phase-matching condition for the second-harmonic uses the second order ($\Lambda = \beta\lambda/2$, $n=2$) meaning that again the interaction with the SH field towards the average acceleration gradient is much degraded.

To my understanding, this all points to a different mechanism that is responsible to the measured electron spectrum. The model depicted in Fig.4e appears to be too rudimentary. Rather than providing more measurements to support this claim, perhaps the authors can perform full 3D simulations (this should be possible with COMSOL, that the authors are already using, by e.g. simulating the generation of SH and then tracking particles).

Rebuttal #4:

I see the authors' claim with Figure S6 (which is a simulation, if I'm not mistaken), and the calculations, but can you prove (at least by answering the questions above) that this is indeed the case in the experiment? It might be that you can indeed get this result from a combination of 515nm+257nm, but I am not convinced yet that this is the case. I suggest to look at particle-tracking simulations (please include in the SI) and see if e.g. more of the decelerating particles "crash" into the structure, leaving a higher number of accelerated particles to reach the energy analyzer. The angle of divergence might also affect the energy analyzer in experiment, by misinterpreting high angles with energy shifts (what aperture did you use?) or missing the energy analyzer altogether.

Answer: Yes, Figure S6 (S4 in revised version) is a result from a combination of fundamental field + SH field. In our experimental measurements, no apertures were used, the electron beam collection angle is larger than 100 m rad.

We have confirmed our data used single nano-wire in which no electrons can “crash” in the samples to impact the spectroscopic analysis [*Nano Letters* 2021, 21, 10238].

In the authors’ 2021 Nano Letters, they used a thin (nanowire) sample, which cannot be compared to the large structure here, especially not in terms of electrons deflecting into the – in the present case – spatially extended structure (nanometers vs. tens of um for the structure and probably some mm for the entire Ag chip – please also provide the total dimensions of the sample, not just the interaction length). I suggest again as in the previous round of review, and my response #3 above, to add particle tracking simulations.

Rebuttal #5:

(2) Damage threshold of Silver in the experiment.

...

...

In experiment, we can measure laser power at a position close to laser port of UTEM chamber.

The relation of laser peak intensity with laser power can be written as $I =$

$\frac{P}{r_{\text{repetition}} \cdot T_{\text{duration}} \cdot \pi(d/2)^2}$, where P is the measured laser power, $r_{\text{repetition}}$ is repetition rate of laser,

T_{duration} is laser pulse duration, d is laser spot diameter. In other hand, peak intensity could also

be written as $I = \frac{1}{2} \epsilon c n E^2$, where ϵ is dielectric constant, c is light speed, n is refractive index.

Therefore, the peak optical field is $E = \sqrt{2I/\epsilon c n}$. Taking the parameters $P=100\text{mW}$,

$r_{\text{repetition}}=200\text{kHz}$, $T_{\text{duration}}=200\text{fs}$, $d=50\mu\text{m}$, $\epsilon=\epsilon_0$, $n=1$, we have $I= 1.27 \times 10^{11} \text{ W/cm}^2$, and $E=0.978 \text{ GV/m}$.

What are the losses at the UTEM port window? Were they taken into account in $P=100\text{mW}$ (is P already in vacuum?) This should be low, of course, but it depends on the type of window and coating you installed – could reduce your values by some 10%.

Rebuttal #6:

How did the authors deduce an acceleration to 2.4 keV? It's most important to be very careful here. Figure 5c shows this, but I would like to see a close-up on that acceleration tail. What is the noise level of this measurement? Can you assert that you are above the noise level and, that this feature is indeed a “real” measurement and not an effect of aberrations through the spectrometer (e.g. high-angle spatial aberrations translate to energy)? A corresponding measurement of the spectrum super-imposed on this curve, with laser off but otherwise exact same parameters, would help quite a lot here.

Answer: 1. A close-up of acceleration tail is shown in below and Fig. 4c in the revised manuscript. It is shown that clear signal can be seen at 2.4 keV above the noise level. Noise level of this measurement is 31 counts (standard deviation).

2. The highest angle the transverse deflection angle can reach to 4.8 milliradians (absorbing 1000 photons). It is calculated by $\theta \approx \arctan \frac{n\hbar k}{p_0}$, where p_0 is initial momentum of electron, $\hbar k$ is the momentum of photon, n is the photon number. For conventional EELS experiments, the zero-loss peak contains electrons with scattering angle from zero to more than one hundred milliradians [Egerton R 1996 *Electron Energy Loss Spectroscopy in the Electron Microscope*]. This fact indicates that high scattering angle electrons can be also correctly characterized by EELS. In addition, electrons with several keV can be well characterized by Gatan imaging filter [<https://live-eels.pantheon.io/products/gatan-imaging-filter-gif>].

3. A relevant measurement of the spectrum at $t = -6.7$ picosecond delay is shown in following figures, which is similar with the laser-off result in Fig. 2a in the revised manuscripts.

Thank you for adding the tail of the distribution to the manuscript, to assess the noise level. With the second figure provided in the rebuttal, lowest here at -6.7 ps, I assume this was taken with the same impact parameter as in the original experiment. In this case, I presume you can explain the decay from 0-0.3 keV as a “standard” EELS spectrum of Silver. What is the glitch at $E = -0.3$ keV?

More importantly, per your calculation, the deflection angle is 4.8 mrad. For an interaction length of only 7.16 μm , the deflection into the structure is $h = \tan(4.8 \text{ mrad}) * 7.16 \mu\text{m} = 34 \text{ nm}$. But your impact factor is 22 nm (in this sense, I do not understand your new inset Fig.4d). Now the question is not the nanophotonic interaction length of 7.16 μm , but the entire sample interaction length. Please provide the full dimensions and entire geometry of the sample, preferably with an image: you do not mention that the bowtie is fabricated on a mesa, so I can assume you have a $\sim 1 \text{ mm}$ silver chip and you placed the bowtie somewhere. Suppose it's in the center, then you have roughly $h = \tan(4.8 \text{ mrad}) * 0.5 \text{ mm} = 2.4 \mu\text{m}$, obviously much larger than the 22 nm impact factor, which would severely distort your final measured spectrum. Please convince me that this is not an issue and does not affect your conclusions (again, full 3D particle tracking simulations with the correct fundamental and SH fields would satisfy me).

Reviewer #3 (Remarks to the Author):

The authors have made substantial changes to the manuscript, which have greatly improved its clarity and readability. I find that my points have been on the whole adequately addressed and I believe the paper is suitable for publication in Nature Communications. However, I would like to point out a few additional points for the authors and editor to consider.

My comment pertains to use of the term “gradient”. There are various conventions for use of this term in the literature. In electron photoinjectors, the maximum cathode field is sometimes called the gradient even though the electron dynamics yield a varying field observed over acceleration from emission to relativistic speed. It is also not uncommon for gradient to be used in reference to the maximal longitudinal electric field intensity in advanced acceleration schemes. In contradiction with this practice, in Ref [17], the axial field is reported as 1.8 GV/m so as to distinguish it from an “average gradient” of 0.850 GV/m given by the measured energy gain of electrons. This variation of nomenclature leads to two observations regarding the present manuscript:

1. The comment on line 46 that “acceleration gradient at the scale of GeV/m has not been realized by DLA to date” is questionable in light of the above example. I would suggest that since the gradient reported by Zheng et al does not exceed 1 GeV/m, this statement does not help motivate the paper and could be removed in favor of emphasizing excitation efficiency, which is substantially higher for the MLA than for DLA.

2. The term “average gradient” as used by Zheng et al, and in their reply to my earlier point #13, appears to mean the average over the energy distribution of the electrons. Thus, for a symmetric energy spectrum (i.e. energy modulation) the average gradient by this definition would be zero. I don't see this as a particularly useful metric. However, I would recommend that the authors explicitly explain within the text that they mean the average over the energy distribution, rather than the more typical interpretation as the average over the length of the structure as seen by a captured or maximally accelerated subset of the total electron population.

Response to Reviewers' Comments

Reviewer #1 (Remarks to the Author):

This reviewer provided confidential remarks to the editor recommending publication.

Reviewer #2 (Remarks to the Author):

Please see pdf attached.

Rebuttal #1:

The authors have added a measurement of laser-off to Fig.2a, but it raises questions. Here is the new Fig.2a from the manuscript:

What is the meaning of the one-sided glitch at the 10^2 counts line, from $x=0$ towards $\sim x=0.2$?

Answer: The laser-off spectrum is an EELS of MLA. Some electrons loss energy and lie in the range from $x=0$ towards ~ 0.2 , because of the interaction between electron beam with silver nanostructures, i.e., a typical laser-off spectrum commonly appears in experimental measurements, which is in connection with excitation of the surface and bulk plasmons, as well as inner shell electrons.

We add a sentence in the caption of Fig. 2 as follow:

The counts from 2 towards 250 eV in the laser-off spectrum arise from the interaction with silver nanostructure without laser excitation.

Rebuttal #2:

This is the core result of this work: showing a nanophotonic antenna interact with an electron.

The authors have two claims here:

(1) Nonlinear effects distort the spectrum towards energy-gain.

I understand from the authors that they assume their “fundamental” on the sample (i.e. 515nm light) goes through a nonlinear process and generates its second-harmonic (257nm). These two interact and are responsible for the energy-spectrum distortion of the electrons. I see some issues with this claim, which to me makes this claim tough to accept:

What is the efficiency of the SH-generation? What would be the calculated field amplitudes (in GV/m) for each?

Answer: The efficiency of SHG in this bowtie structure depends mainly on the intensity of laser. In our theoretical simulation, the weighted average fundamental acceleration field is 0.156 GV/m (corresponding $|g_1|=233$), SH acceleration field is 0.0312 GV/m (corresponding $|g_2|=23.3$). Therefore, the SH near field is one-fifth of fundamental field. Also see the answer for question 6 of reviewer#1.

The authors state here $|g_2|/|g_1| = 0.1$, but in the caption of Figure 4 $|g_2|/|g_1| = 0.2$. In Fig.S5, the authors pick $|g_1|=300$ and $|g_2|=26$, so $|g_2|/|g_1| = 0.087$. Please explain and then correct these discrepancies.

Could it be that the authors confuse again the acceleration gradient G (units of eV/m) with field (units of V/m)? How does the “weighted average fundamental acceleration field” G_1 of 0.156 GV/m and the “SH acceleration field” G_2 of 0.0312 GV/m, relate to the ultimate average acceleration gradient result of 0.335 GeV/m?

Assuming the authors meant that the contribution to the acceleration gradient from the fundamental is $G_1=0.156$ GeV/m and from the second harmonic $G_2=0.0312$ GeV/m, how is this weighted average calculated? It seems like the contribution of the SH is almost an order of magnitude less than the fundamental – I find it difficult to agree that the SH has such a large impact on the electron spectrum. Was Fig.4e calculated with any impact factor? Or is it a 1D simulation? This should be noted in the text.

Answer: 1. Here $|g_2|/|g_1| = 0.1$ is the ratio of weighted averages. In the caption of Fig. 4, it is just an example of the electron distribution in each energy state at low field intensity. In Fig. S5b, to make it clearly shows the electron distribution at high field intensity, we pick the spectrum at $|g_1|=300$ and $|g_2|=26$ as an example of Fig. S5a (corresponding the red dash line).

2. Yes, it should be replaced by GeV/m. The “weighted average acceleration” is a value in quantum simulation to measure the mean of electron energy-gains. The ultimate average acceleration gradient result of 0.335 GeV/m is an experimental value, related to the highest energy-gain electrons (2.4 keV).

3. To clearly explain the origin of “weighted average”, an explanation is shown as follow: Since we assume the intensity of second harmonic field is quadratic increase depend on the fundamental field, the relation of $|g_1|$ with $|g_2|$ can be written as $|g_2|=k \cdot |g_1|^2$, where $k=2.86 \times 10^{-4}$ in the simulation (see low panel of Fig. S5a). The probabilities of electron in different near-fields are represented by distribution function $D(|g|)$, which satisfied $\int D(|g|)d|g| = 1$ (see SI 2.3). We assume $D(|g|)$ can be expressed as $D(|g|) = a(|g|_{\max} - |g|)$, a is constant coefficient (see right panel of Fig. S5a). The weighted average coupling parameter can be expressed as

$$|g|_{\text{ave}} = \int |g| \cdot D(|g|)d|g|.$$

In our simulation, $|g_1|$ ranges from 0 to 700. Thus, weighted average $|g_1|$ equals to 233 (corresponding $|G_1|=0.156$ GV/m, according the equation (2) in main text, i.e., $|g| = \frac{N\Lambda}{2\hbar\omega} |G|$), weighted average $|g_2|= 23.3$ (corresponding $|G_2|=0.0312$ GV/m).

4. Due to the quadratic increase relation, SH field only distributes in areas with strong fundamental field. Therefore, the SH contributes mainly to the high energy-gain electrons, yielding the notable asymmetry in high energy region of spectra. There has no any impact fact on Fig. 4e (not a 1D simulation), as the answer 2 to review #1 in previous version.

Rebuttal #3:

Continuing the point about G1 and G2 from the previous point, now the decay lengths are half of each other, plus the phase-matching condition for the second-harmonic uses the second order ($\Lambda = \beta\lambda/2$, $n=2$) meaning that again the interaction with the SH field towards the average acceleration gradient is much degraded.

To my understanding, this all points to a different mechanism that is responsible to the measured electron spectrum. The model depicted in Fig.4e appears to be too rudimentary. Rather than providing more measurements to support this claim, perhaps the authors can perform full 3D simulations (this should be possible with COMSOL, that the authors are already using, by e.g. simulating the generation of SH and then tracking particles).

Answer: The simulation is carried out in particle tracing module (COMSOL Multiphysics 6.0). The time-dependent electromagnetic field is calculated in radio frequency module. In the simulation, ten thousand electrons with the impact factor from 0 to 45 nm are taken into our calculation and the electron pulse temporal width is set to 1.75 times to laser pulse width, the same to the experimental parameters. Both the time-dependent acceleration (eE_z) and time-dependent deflection force ($eE_x + ev_e B_y$) are considered for the electron grazing over the 20-period-MLA. In the case of coexisting

ω and 2ω fields, a light beam with 515 nm wavelength and another light beam with 257.5nm are incident to MLA. Only electrons without hitting the MLA are counted in the final electron energy spectra. The particle tracing simulation results shown in below (also shown in Fig. 4e and demonstrated in Method parts).

Rebuttal #4:

In the authors' 2021 Nano Letters, they used a thin (nanowire) sample, which cannot be compared to the large structure here, especially not in terms of electrons deflecting into the – in the present case – spatially extended structure (nanometers vs. tens of μm for the structure and probably some mm for the entire Ag chip – please also provide the total dimensions of the sample, not just the interaction length). I suggest again as in the previous round of review, and my response #3 above, to add particle tracking simulations.

Answer: Two SEM images of our MLA are shown below. The silver nanostructures are severely oxidative and damaged by the fs-laser upon high power illumination. MLA is made from a silver plate with a thickness of only $25\mu\text{m}$. The up and down parts of nanostructure had been removed by ion beam. The depth of grooves is more than one micrometer.

Rebuttal #5:

What are the losses at the UTEM port window? Were they taken into account in $P=100\text{mW}$ (is P already in vacuum?) This should be low, of course, but it depends on the type of window and coating you installed – could reduce your values by some 10%.

Answer: We had already considered the losses at the UTEM port window. The port window is made of CaF_2 with the transmission rate of $\sim 87\%$. In order to get the $P=100\text{mW}$, the laser power is set to 115mW before enter port window (as measured at air).

Rebuttal #6:

Thank you for adding the tail of the distribution to the manuscript, to assess the noise level. With the second figure provided in the rebuttal, lowest here at -6.7 ps , I assume

this was taken with the same impact parameter as in the original experiment. In this case, I presume you can explain the decay from 0-0.3 keV as a “standard” EELS spectrum of Silver. What is the glitch at E=0.3 keV?

More importantly, per your calculation, the deflection angle is 4.8mrad. For an interaction length of only 7.16 um, the deflection into the structure is $h=\tan(4.8\text{mrad})\cdot 7.16\text{um} = 34\text{nm}$. But your impact factor is 22nm (in this sense, I do not understand your new inset Fig.4d). Now the question is not the nanophotonic interaction length of 7.16um, but the entire sample interaction length. Please provide the full dimensions and entire geometry of the sample, preferably with an image: you do not mention that the bowtie is fabricated on a mesa, so I can assume you have a ~1mm silver chip and you placed the bowtie somewhere. Suppose it’s in the center, then you have roughly $h=\tan(4.8\text{mrad})\cdot 0.5\text{mm} = 2.4\text{um}$, obviously much larger than the 22nm impact factor, which would severely distort your final measured spectrum. Please convince me that this is not an issue and does not affect your conclusions (again, full 3D particle tracking simulations with the correct fundamental and SH fields would satisfy me)

Answer: 1. The glitch at E=0.3 keV is the X-ray spikes in electron spectrometer. It is a common feature appearing in EELS spectra.

2. We performed a calculation based on our experimental data to give the clear results. The deflection is an incremental process, it is not reasonable using $h=\tan(4.8\text{mrad})\cdot 7.16\text{um} = 34\text{nm}$. The deflection should be calculated in our case

$$h = \int_0^L \frac{n \hbar k}{L p_0} z dz = \frac{n \hbar k L}{p_0^2} = \tan \theta \cdot \frac{L}{2},$$

where L is the interaction length, p_0 is initial momentum of electron, $\hbar k$ is the momentum of photon, n/L is the absorbed photon number per interaction length. For the interaction length of 7.16 um and highest deflection angle 4.8mrad, we get **$h=17\text{nm}$** . It must be noted that the electron beam has a diameter of ~45 nm (see the first paragraph of Results in main text). Impact factor = 0 nm means the distance between the center of electron beam to the MLA is 0 nm, it doesn’t mean no electrons can pass through MLA.

3. The images of entire geometry of the sample are shown in the answer of question #4. The max deflection $h=17\text{nm}+\tan(4.8\text{mrad})\cdot 10\text{um}=65\text{nm}$ is much less than the depth of groove, which have specially dug in our nanostructure for UTEM observations.

4. In order to clearly show that the asymmetry is due to the generation of second harmonic field, not due to some electron hit the nanostructure, another simulation is carried out. All electrons have an impact factor of 20 nm, the temporal width of laser pulse is much large than electron pulse, a close-up of electrons pass last tooth of MLA

is shown in below. There have no any electrons hit MLA both in the case of only fundamental fields and coexisting SH fields.

The final electron energy distribution is shown in below, which clearly revealed the existence of SH field will result in the appearance of spectrum asymmetry.

If we set the impact factor =5 nm, some electrons will hit the nanostructure (below left), but the final electron energy spectrum is symmetric (below right). This fact indicates that some electrons hit nanostructure cannot result in spectrum asymmetric.

Reviewer #3 (Remarks to the Author):

The authors have made substantial changes to the manuscript, which have greatly improved its clarity and readability. I find that my points have been on the whole adequately addressed and I believe the paper is suitable for publication in Nature Communications. However, I would like to point out a few additional points for the authors and editor to consider.

Answer: Thanks for the positive comments.

My comment pertains to use of the term “gradient”. There are various conventions for use of this term in the literature. In electron photoinjectors, the maximum cathode field is sometimes called the gradient even though the electron dynamics yield a varying field observed over acceleration from emission to relativistic speed. It is also not uncommon for gradient to be used in reference to the maximal longitudinal electric field intensity in advanced acceleration schemes. In contradiction with this practice, in Ref [17], the axial field is reported as 1.8 GV/m so as to distinguish it from an “average gradient” of 0.850 GV/m given by the measured energy gain of electrons. This variation of nomenclature leads to two observations regarding the present manuscript:

1. The comment on line 46 that “acceleration gradient at the scale of GeV/m has not been realized by DLA to date” is questionable in light of the above example. I would suggest that since the gradient reported by Zheng et al does not exceed 1 GeV/m, this statement does not help motivate the paper and could be removed in favor of emphasizing excitation efficiency, which is substantially higher for the MLA than for DLA.

Answer: Thanks for the suggestion. It has been revised as follows:

Though the driving fields have reached 9 GV/m for the DLA^{16,17}, yielding the highest average acceleration gradient to date, the rate of average acceleration gradient for incident laser field is pretty low. The exploration of high acceleration gradient accelerator and new laser-driven accelerator designs continues.

2. The term “average gradient” as used by Zheng et al, and in their reply to my earlier point #13, appears to mean the average over the energy distribution of the electrons. Thus, for a symmetric energy spectrum (i.e. energy modulation) the average gradient by this definition would be zero. I don't see this as a particularly useful metric. However, I would recommend that the authors explicitly explain within the text that they mean the average over the energy distribution, rather than the more typical interpretation as

the average over the length of the structure as seen by a captured or maximally accelerated subset of the total electron population.

Answer: Since the term “average gradient” is widely interpreted as “average over the length of the structure” in previous literatures, to avoid the potential misunderstanding of “average gradient”, we only adopt this mean in revised manuscript. The mean of “average over the energy distribution” had been deleted.

REVIEWERS' COMMENTS

Reviewer #2 (Remarks to the Author):

I thank the authors for performing the particle tracking simulations. They are indeed helpful. Please add them to the supplement so that the article reader can also see them. Once this is done, it is a pleasure for me to recommend publication of this work in Nature Communications.

Reviewer #3 (Remarks to the Author):

I have reviewed the revised manuscript and find that the points from my prior reviews have been adequately addressed. I recommend the paper for publication.

Response to Reviewers' Comments

Reviewer #2 (Remarks to the Author):

I thank the authors for performing the particle tracking simulations. They are indeed helpful. Please add them to the supplement so that the article reader can also see them. Once this is done, it is a pleasure for me to recommend publication of this work in Nature Communications.

Answer: The particle tracing simulation results had been shown in Fig. 4e and Fig. S4. We add following sentences in the page 4:

To clearly show that the asymmetry is not due to partial electrons hitting the nanostructure and uncollected by electron spectrometer, another simulation is carried out. When the incident electron beam is very close to MLA (small impact factor), some electrons will hit MLA (Fig. S4), but the final electron energy spectrum is symmetric.

Reviewer #3 (Remarks to the Author):

I have reviewed the revised manuscript and find that the points from my prior reviews have been adequately addressed. I recommend the paper for publication.

Answer: Thanks for the positive comments.